# Understanding the Geospatial Reasoning Capabilities of LLMs: A Trajectory Recovery Perspective

## Abstract

We explore the geospatial reasoning capabilities of Large Language Models (LLMs), specifically, whether LLMs can read road network maps and perform navigation. We frame trajectory recovery as a proxy task, which requires models to reconstruct masked GPS traces, and introduce GLOBALTRACE, a dataset with over 4,000 real-world trajectories across diverse regions and transportation modes. Using road network as context, our prompting framework enables LLMs to generate valid paths without accessing any external navigation tools. Experiments show that LLMs outperform off-the-shelf baselines and specialized trajectory recovery models, with strong zero-shot generalization. Fine-grained analysis shows that LLMs have strong comprehension of the road network and coordinate systems, but also pose systematic biases with respect to regions and transportation modes. Finally, we demonstrate how LLMs can enhance navigation experiences by reasoning over maps in flexible ways to incorporate user preferences.

## 1 Introduction

Large Language Models (LLMs) are increasingly recognized as general-purpose systems, showing strong performance across domains ranging from mathematics and coding to vision and robotics. An emerging yet underexplored question is whether these models possess geospatial understanding, the ability to reason about maps, paths, and spatial relationships. Such capabilities are fundamental to many real-world applications, e.g., autonomous vehicle navigation, logistics, and urban planning. While prior work has studied LLMs in contexts such as geographic knowledge retrieval (Manvi et al., 2024a;b) and map-based multiple-choice question answering (Dihan et al., 2025), the ability of LLMs to read road networks and plan paths has not been systematically evaluated.

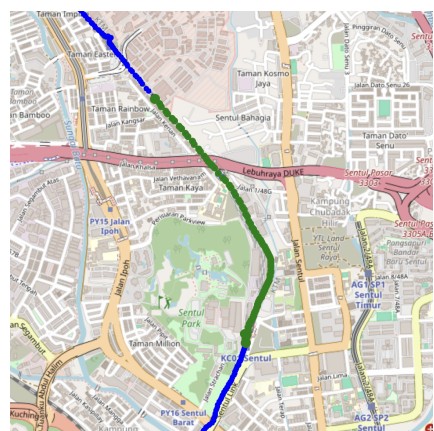

Figure 1: An example of the trajectory recovery task of a sample in GLOBAL-TRACE. The masked segment to recover is in green, which is part of a trajectory shown in blue.

We investigate whether LLMs can perform navigation through the trajectory recovery task: reconstructing masked segments of GPS traces from the road network context, to bypass the restriction of relying on shortest path-type of ground truth which may not reflect human navigation pattern in practice (Golledge, 1995; Duckham & Kulik, 2003). To facilitate this evaluation, we introduce GLOBALTRACE, a dataset of over 4,000 real-world trajectories spanning multiple continents and diverse transportation modes, collected from public traces on OpenStreetMap (OpenStreetMap contributors, 2017). Unlike existing datasets, GLOBALTRACE covers multiple activities, i.e., transportation modes (e.g., driving, cycling, walking, and hiking), and geographic regions, enabling a comprehensive evaluation of model generalization. Our dataset is framed in away that is harder than the traditional point-wise trajectory recovery task (Newson & Krumm, 2009; Song et al., 2017; Si et al.,

2024), and closer to the higher-level navigation problem. As Figure 1 shows, to successfully reconstruct a masked segment of a trajectory, a model must be able to (1) understand the context of travel (e.g., mode of transportation, speed, and direction) as well as the constraints of the road network (e.g., one-way road and road connectivity) to select an appropriate route and (2) generate smooth coordinates following the route.

Traditionally, the task of trajectory recovery is tackled using either probabilistic models and map-matching techniques (Newson & Krumm, 2009; Huang et al., 2018; Yin et al., 2018), such as Hidden Markov Models (HMM), or interpolation heuristics (Zheng et al., 2008), to reconstruct missing points from noisy or sparse GPS traces. While effective in constrained settings, these approaches are limited in capturing long-range dependencies, or adapting to complex moving patterns. Following the pre-training trend after Transformer (Vaswani et al., 2017) is introduced, there is also effort to build foundation trajectory models (Liang et al., 2025; Lin et al., 2024; Si et al., 2024; Yu et al., 2025b) to encode geospatial knowledge from large amount of training data. This approach relies on training using specialized in-domain trajectory data and struggles to generalize to unseen regions.

To address these issues, we explore the generalization capability of LLMs and present a prompting-based framework that allows LLMs to reconstruct valid trajectories without reliance on external navigation tools. Our system reasons over the available road network data to perform reconstruction without the need of additional in-domain training. Our experiments compare LLMs against both off-the-shelf navigation systems and specialized trajectory recovery models. Results show that state-of-the-art LLMs achieve superior performance, with strong zero-shot generalization across regions and activity types. Fine-grained analysis highlights their ability to plan over complex road networks, adhere to geometric constraints, and generate realistic coordinates, while ablation studies reveal that compact topological representations with directional cues are the most effective input format.

This combination of geospatial reasoning with LLMs' semantic strengths to understand flexible user queries opens new opportunities for navigation. For example, beyond recovering trajectories, an LLM could generate a route that is not only efficient but also tailored to a user's preferences, such as scenic views, safety, or avoiding noisy streets—by integrating contextual knowledge already embedded in the model. Such preference-aware navigation lies beyond the scope of traditional systems, but emerges naturally when map reasoning is combined with LLMs' flexible language and reasoning capabilities.

Our contributions are threefold:

- We introduce **GLOBALTRACE**, a benchmark of over 4,000 real-world trajectories across regions and transportation modes to benchmark models' geospatial reasoning capabilities.

- We propose a prompting-based framework that elicits LLMs' capability to perform trajectory recovery directly from the road network context, without external navigation tools.

- We provide extensive evaluation and analysis, showing that LLMs outperform specialized trajectory recovery models, generalize zero-shot, exhibit systematic regional and activity biases, and can serve as a foundation for preference-based navigation. We release the reproducible code for the experiment and our system together with the GLOBALTRACE dataset at `https://anonymous.4open.science/r/llm_traj_rec-5F5B/`.

## 2 RELATED WORK

We covered a summary of existing works on trajectory recovery in the last section. Below, we outline the literature that studies the geospatial understanding capabilities of LLMs. A full discussion on the related works is included in Appendix A.

Recent studies have probed whether large language models possess an implicit grasp of geographic space and spatial reasoning. Gurnee & Tegmark (2024) show that LLMs are able to map coordinates to corresponding regions. Manvi et al. (2024b) show that LLMs embed substantial geospatial knowledge (e.g., population density and mean income), but simply feeding geographic coordinates into prompts yields poor results on those questions. The authors augment coordinate inputs with auxiliary map data (from OpenStreetMap), such as point-of-interest (POIs) around the coordinates, and achieve large improvement over naive baselines. Manvi et al. (2024a) further extend this analysis, showing that LLMs exhibit systematic geographic biases (e.g., against lower-socioeconomic status

regions). MapEval (Dihan et al., 2025) is a benchmark of 700 multiple-choice, map-grounded questions (textual/API/visual) across 180 cities and 54 countries. Across 30 models tested, none exceeds 67% accuracy and all lag behind human performance, with particular difficulties in distance/direction inference, route planning, and visual map understanding. Unlike previous task formulations of knowledge extraction, bias diagnostic, and multiple-choice question answering, we tackle trajectory recovery as a path reconstruction problem and investigate whether general-purpose LLMs can perform this in a zero-shot manner, without in-domain pretraining or extra supervision.

## 3 GLOBALTRACE

To enable comprehensive benchmarking, we curate a diverse dataset named GLOBALTRACE covering multiple modes of transportation and geographic regions. The dataset is derived from publicly available user traces on OpenStreetMap, collected via its official API (Overpass API, 2025) across a predefined set of regions and cities (detailed in Appendix B). As these traces are voluntarily shared by contributors and contain no personally identifiable information (PII), their use raises no privacy concerns. We further constrain all trajectories to lie within a target length range of 500 m to 30,000 m to provide sufficient context for meaningful reconstruction. To lessen the risk of data contamination, we only consider traces from 2024 onward. Each trace is annotated with an activity type (i.e., mode of transportation) using keyword matching over the trace name and description, with an LLM-based classifier serving as a fallback to allow semantic-based mapping and for cases such as non-English text. For every trace, we generate two masked variants by masking consecutive points: one with a small gap and another with a large gap, providing different difficulty levels. To facilitate different experiment settings, we partition the dataset into training, development, and test splits, stratified to balance both activity types and regional coverage. We provide the geographic coverage, activity type distribution, and route length statistics of GLOBALTRACE in Figure 2.

**Problem Statement** Given a partial trajectory $T = \langle p_1, p_2, \ldots, p_{s-1}, p_s, p_e, p_{e+1}, \ldots, p_{|T|} \rangle$, where each point $p_i$ is represented by its coordinates (latitude and longitude), and there is a consecutive segment $G = \langle p_1^*, p_2^*, \ldots, p_{|G|}^* \rangle$ between $p_s$ and $p_e$ that is missing (i.e., masked), our *masked trajectory reconstruction* task aims to generate a series of points $R = \langle \hat{p}_1, \hat{p}_2, \ldots, \hat{p}_{|R|} \rangle$ that minimizes the *deviation* of $R$ from $G$. In Section 5, we will detail how to measure the deviation.

Our masked trajectory reconstruction task departs significantly from two common setups in the literature. First, it is not a map-matching problem (Yin et al., 2018; Ren et al., 2021) where the goal is to align noisy or low-quality GPS points to the underlying road network. In our problem, the coordinates to be recovered (i.e., $G$) are entirely withheld rather than available in degraded form. Second, our trajectory reconstruction problem is more challenging than point-wise recovery tasks (Si et al., 2024; Zhu et al., 2024), where single (or a few) missing points are inferred from its immediate neighbors. Instead, we remove a contiguous segment of the trajectory, spanning hundreds to thousands of meters. A model must therefore reason over global context (movement context before and after the gap, activity type, and road network topology) to generate coherent continuation. This long-horizon prediction makes our problem substantially more challenging , and better reflects the complexity of real-world mobility data.

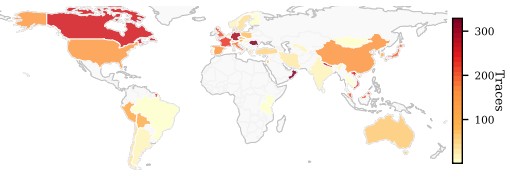

| | **Masked (km)** | | | **Full trajectory (km)** | | |
|---|---|---|---|---|---|---|
| **Strategy** | Mean | Min | Max | Mean | Min | Max |
| Small gap | 0.3 | 0.2 | 0.5 | 8.6 | 0.5 | 29.9 |
| Large gap | 1.8 | 0.5 | 2.9 | 10.2 | 1.1 | 29.9 |

| **Activity** | **Dev** | **Test** | **Train** | **Total** |
|---|---|---|---|---|
| Hiking | 63 | 151 | 1188 | 1402 |
| Driving | 228 | 231 | 491 | 950 |
| Walking | 73 | 48 | 556 | 677 |
| Cycling | 127 | 12 | 380 | 519 |
| Bus | 68 | 87 | 77 | 232 |
| Train | 29 | 35 | 156 | 220 |
| Boat | 4 | 2 | 74 | 80 |
| Flying | 4 | 3 | 8 | 15 |
| **Total** | **596** | **569** | **2930** | **4095** |

Figure 2: GLOBALTRACE dataset overview: (a) geographic coverage and trajectory-length statistics, (b) activity type distribution.

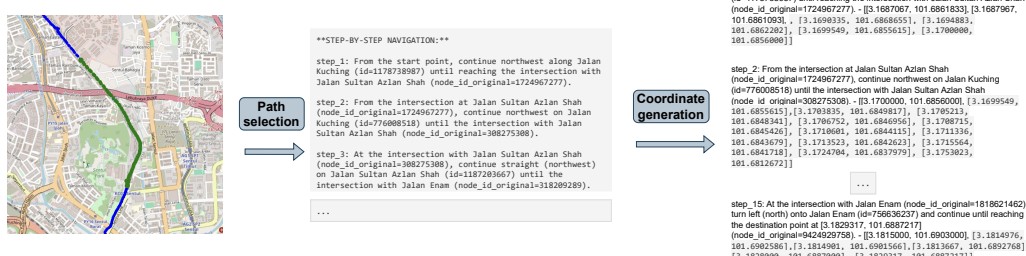

Figure 3: Two-stage LLM trajectory reconstruction.

## 4 METHODS

Traditional Transformer-based methods in trajectory reconstruction require a large amount of training data (e.g., 2.4M trajectories (Zhu et al., 2024)). Moreover, such models fail to generalize beyond the region covered by their training data (typically just a city). By contrast, we propose a framework that enables LLMs to perform navigation-style reasoning without relying on external routing engines or domain-specific training.

A naive solution is to prompt an LLM with the partial trajectory to be recovered, $T$, and the road network context of the region covering $T$. Empirically, we find that this naive solution fails to elicit the geospatial reasoning capabilities of LLMs, due to a large amount of unorganized data being fed to the LLM. We detail such alternative solutions in Section 6.2.

To elicit the geospatial reasoning capabilities of LLMs, we adopt a two-stage, schema-constrained design: (1) **Path selection** over road and intersection IDs; and (2) **Coordinate generation** grounded by road geometries, as illustrated in Figure 3. This two-stage design helps evaluates whether LLMs can reconstruct trajectories by reasoning over road network topology rather than delegating to external navigation engines.

Our two-stage approach decouples the two distinct goals and allow fine-grained qualitative analysis: in the first stage, we exploit LLMs' strong reasoning abilities to generate a verbose plan, while in the second stage, we instruct the models to strictly adhere to the plan by grounding against road geometries to generate precise coordinates (i.e., points in $R$). Unlike prior work that either performs map-matching from noisy GPS points or predicts a single missing location given surrounding context, our formulation demands recovering an entire long masked segment of trajectory. This setup is significantly harder, as the model must maintain consistency across multiple steps, align with motion summaries, and adhere to road network constraints simultaneously.

**Stage 1: Path Selection** This stage evaluates the ability of LLMs to read the road network surrounding the masked segment and perform planning to craft an appropriate path connecting the start and end points $p_s$ and $p_e$ with respect to the trajectory context. We provide the following context to an LLM: (1) **Masked segment information:** coordinates of $p_s$ and $p_e$, activity type, and masked segment length. (2) **Trajectory context summaries:** timing, speed, and heading narratives before and after the masked segment, which provide more context to help models understand the movement. (3) **Road network slice:** a road network enclosed by a bounding box surrounding the masked segment. Each road in the road network has ID, name, type, cardinal direction, and the list of roads that it connects to along with the corresponding intersection ID. coordinates, and precomputed bearing to destination. We detailed the road network construction in Appendix C and compare different ways to represent it in our ablation study (Section 6.2). We enforce further requirements for the navigation steps to make sure that LLMs produce unambiguous description with clear anchors (full prompt in Appendix D).

**Stage 2: Coordinate Generation** Once a valid step-by-step plan is produced, the second stage requests explicit coordinates (points in $R$) for each step. We perform this iteratively for each step, grounded against the road network to ensure models generate valid coordinates that adhere to the roads. To be able to successfully generate coordinates, the LLM must (1) be able to align the

generation to the provided road geometries, and (2) generate coordinates in the order that respect the moving direction in the generated step descriptions. Please refer to Appendix D for an example of the prompt and response.

## 5 EXPERIMENTAL SETUP

**Models Evaluated**  We consider three classes of models to benchmark against GLOBALTRACE. The settings and parameters are detailed in Appendix E.

1. **Baselines.** Traditional baselines for trajectory recovery. We consider: Linear interpolation ("**Linear**"), and Linear interpolation + HMM map-matching ("**Liner+HMM**"), following Yu et al. (2025b). We also consider a popular navigation system **Google Maps**. Although not designed for trajectory recovery, we convert the path generated by Google Maps Direction API (as polylines) into coordinates by decoding the Google's Encoded Polyline Algorithm Format (Google Maps Platform) for comparison with other methods.

2. **Pre-trained models.** We compare with a strong pre-trained Transformer model for trajectory recovery: TrajFM (Lin et al., 2024). The model was pre-trained using the Didi dataset (Didi-Chuxing, 2018) containing vehicle trajectories in two cities in China. We fine-tuned the model on our GLOBALTRACE training set.

3. **LLM models.** We consider strong models from different model families, including three proprietary models: GPT-4.1, GPT-4.1-mini, and Claude 4 Sonnet, and four open-weight models: Deepseek V3, Llama 4 Maverick, Qwen3-235B, and Qwen3-30B.

**Metrics**  We focus on two evaluation metrics defined based on $G = \langle p_1^*, p_2^*, \ldots, p_{|G|}^* \rangle$, the ground-truth points of a masked segment, and $R = \langle \hat{p}_1, \hat{p}_2, \ldots, \hat{p}_{|R|} \rangle$, the reconstructed points.

*Mean Absolute Error* (MAE; $\downarrow$), a commonly used metric in the trajectory recovery literature (Zhu et al., 2024), measures the average distance deviation (the the Haversine distance, denoted by $d(\cdot, \cdot)$) between the original trajectory points and their respective closest points on the reconstructed trajectory, normalized by the total masked segment length $L_G$.

$$\text{MAE}_{G \to R} = \frac{1}{|G| \cdot L_G} \sum_{i=1}^{|G|} \min_{j=1}^{|R|} d(p_i^*, \hat{p}_j) \cdot 100. \tag{1}$$

To penalize spurious, off-route reconstruction, we also compute $\text{MAE}_{R \to G}$ that measures the average distance deviation between the reconstructed trajectory points and their respective closest points on the ground-truth trajectory, expressed as a percentage of the reconstructed trajectory length $L_R$. We combine $\text{MAE}_{G \to R}$ and $\text{MAE}_{R \to G}$ to calculate the harmonic mean.

$$\text{MAE}_{F1} = \frac{2 \cdot \text{MAE}_{G \to R} \cdot \text{MAE}_{R \to G}}{\text{MAE}_{G \to R} + \text{MAE}_{R \to G}}. \tag{2}$$

While commonly used, MAE is sub-optimal to measure the path reasoning capabilities of LLMs as it is a point-to-point metric, which can be affected when we have different point density between the ground truth and the generated trajectory. To address this, we propose a point-to-segment metric.

*Point-on-Trajectory* (PoT; $\uparrow$) measures the percentage of original trajectory points that fall within $\tau$ ($\tau = 10$ by default) meters of the reconstructed trajectory. For each ground-truth point, we calculate the minimum distance to the reconstructed trajectory (considering all segments between any two consecutive points). We count the number of points within the $\tau$-meter tolerance. Higher PoT values indicate better ground-truth route coverage. This metric is less sensitive to minor positional errors, focuses on overall route adherence, and penalizes reconstructions that miss significant portions of the original trajectory.

$$\text{PoT}_{G \to R} = \frac{1}{|G|} \sum_{i=1}^{|G|} \mathbb{I}\left( \min_{j=1}^{|R|-1} d_{\text{seg}}(p_i^*, \overline{\hat{p}_j, \hat{p}_{j+1}}) \leq \tau \right) \cdot 100. \tag{3}$$

Here, $\mathbb{I}(\cdot)$ is the indicator function, and $d_{\text{seg}}(\cdot, \cdot)$ is the distance from a point to a line segment. Like before, we also compute $\text{PoT}_{R \to G}$ that measures the percentage of reconstructed trajectory points

that fall within $\tau$ meters of the ground-truth trajectory. The harmonic mean, which we call the symmetric coverage score, is then calculated as:

$$\text{PoT}_{F1} = \frac{2 \cdot \text{PoT}_{G \to R} \cdot \text{PoT}_{R \to G}}{\text{PoT}_{G \to R} + \text{PoT}_{R \to G}}. \tag{4}$$

# 6 MAIN FINDINGS

## 6.1 RQ1: CAN LLMs PERFORM TRAJECTORY RECOVERY?

Table 1 reports the trajectory recovery results on the GLOBALTRACE test set. Overall, we can see that, LLMs achieve strong results, on par with Google Maps, which is the state-of-the-art (SOTA) navigation system. Even Google Maps has a relatively low results for this task, showing that actual user trajectories do not always follow the optimal paths recommended by professional navigation service, especially for exploration type activities such as walking or hiking (cf. Figure 6b).

Results of different LLMs tend to fall in similar range of 50-60%, with GPT-4.1 producing the best results, followed by Claude-4-Sonnet. The two smaller variants, GPT-4.1-mini and Qwen-3-30B, have worse results compared to their larger counterparts, indicating that geospatial reasoning capability emerges with scale. The pre-trained model, TrajFM, on the other hand, performs the worst, and most of the time it just draws a straight line in small gap cases, or otherwise generate coordinates in a completely different region. This is expected as we ensure no overlap between the regions in the train/development and the test sets. It shows that generalizing to an unseen cities/region is not a trivial fine-tuning objective and would likely require large amount of training data across different regions in the world. We discuss more on the mismatch between the two metrics and show cases where LLMs perform better than Google Maps in Appendix I, as well as the detailed results for each reconstruction direction ($G \to T$ and $T \to G$) in Appendix H.1.

| | Method | Small gap | | Large gap | | Overall | |
|---|---|---|---|---|---|---|---|
| | | $\text{PoT}_{F1}\uparrow$ | $\text{MAE}_{F1}\downarrow$ | $\text{PoT}_{F1}\uparrow$ | $\text{MAE}_{F1}\downarrow$ | $\text{PoT}_{F1}\uparrow$ | $\text{MAE}_{F1}\downarrow$ |
| *Baseline* | Google Maps | **69.9** | 15.5 | **60.0** | 8.0 | **65.0** | 11.8 |
| | Linear | 65.7 | **4.2** | 25.0 | 5.1 | 45.4 | 4.7 |
| | Linear+HMM | 63.2 | 5.3 | 25.9 | 5.0 | 44.9 | 5.2 |
| *Pre-trained* | TrajFM | 23.6 | 46.5 | 10.2 | 59.3 | 15.3 | 50.5 |
| *LLM* | GPT-4.1 | 68.4 | 7.1 | 58.1 | 3.8 | 63.3 | 5.4 |
| | GPT-4.1-mini | 63.7 | 7.3 | 50.2 | 4.4 | 57.0 | 5.9 |
| | Claude-Sonnet-4 | 63.1 | 8.0 | 53.3 | 4.4 | 58.2 | 6.2 |
| | Llama-4-Maverick | 53.0 | 7.0 | 44.5 | 4.2 | 48.7 | 5.6 |
| | DeepSeek-V3 | 61.7 | 7.7 | 49.8 | 4.5 | 55.8 | 6.1 |
| | Qwen-3-235B | 57.2 | 6.6 | 47.4 | **3.7** | 52.4 | **5.2** |
| | Qwen-3-30B | 58.6 | 7.6 | 45.2 | 4.7 | 51.9 | 6.2 |

Table 1: Reconstruction performance on the GLOBALTRACE test set. Best scores are in **bold**, second-best are underlined.

**Stage-Based Analysis**  As our method consists of two stages, we also perform analyses for each stage to assess their output quality. For path finding, we measure the **path connectivity**, i.e., whether the generated step-by-step navigation can form a valid path connecting the start and end points of the masked trajectory. We extract road segments from the generated step-by-step navigation and cross-check them against the road network graph to verify if all the elements are connected. As shown in Table 2, all tested LLMs show above average scores, with GPT-4.1 being the strongest model (i.e., all road segments in the generated path are connected for 76.2% of the test instances), and Qwen-3-30B the weakest.

We also report the **road network adherence** ("Net. adh.") score, which measures the proportion of the generated roads IDs and intersection IDs strictly presented in the road network. The scores for all LLMs tested are almost perfect, showing almost zero hallucination in generating road IDs and intersection IDs adhering to the provided road network. The **average number of steps** ("Avg. # steps")

are between 3 and 5 for most models, while the **average number of gaps** ("Avg. # gaps") longer than 200 m (which is considered a large gap) between steps is below 0.2, showing smooth transition between navigation steps. These results show that LLMs pose strong path finding capabilities. They can plan and reason over complex and large road network graphs to find valid paths.

For coordinate generation, we evaluate whether the generated coordinates for each step are consistent with the step description. This includes **geometry adherence** (percentage of the generated coordinates presented in the road geometry, "Geo. adh."), and **bearing error** (how far the directions of the generated coordinates deviate from the step descriptions, "Bearing"). To determine the direction of the coordinates for each step, we calculate the bearing using the start and end points for the step, then measure the error with respect to the direction (north, east, south, or west) in the step description. The bearing errors of all models are in the acceptable quadrant (i.e., $< 90°$), with the Qwen and DeepSeek models having slightly larger deviation. The coordinate generation analysis demonstrates LLMs' strong awareness of the GPS coordinate systems, i.e., the models can correctly generate coordinates following the correct direction of the selected road segments.

Overall, the fine-grained quality metrics have high correlation with the overall trajectory recovery performance, i.e., the models score consistently higher (or lower) for both evaluations.

| Model | Path Finding | | | | Coordinate Generation | |
|---|---|---|---|---|---|---|
| | Connectivity (%)↑ | Net. adh. (%)↑ | Avg. # gaps↓ | Avg. # steps | Geo. adh. (%)↑ | Bearing (°)↓ |
| GPT-4.1 | **76.2** | **99.8** | **0.07** | 4.2 | 84.0 | 54.8 |
| GPT-4.1-mini | 67.5 | **99.8** | **0.07** | 3.9 | 80.6 | **54.3** |
| Claude-4-Sonnet | 68.3 | 99.7 | 0.15 | 4.3 | **86.8** | 54.9 |
| Llama-4-Maverick | 64.8 | 99.3 | 0.13 | 5.0 | 76.2 | 55.7 |
| DeepSeek-V3 | 61.6 | **99.8** | 0.14 | 4.0 | 81.7 | 56.4 |
| Qwen-3-235B | 63.8 | 98.6 | 0.27 | 4.7 | 82.3 | 55.8 |
| Qwen-3-30B | 55.2 | 97.8 | 0.38 | 3.7 | 80.2 | 57.3 |

Table 2: Quality analysis for the two-stage LLM-based approach. Best scores are in **bold**, second-best are underlined. The metrics are defined in Appendix G.

## 6.2 RQ2: WHAT ROAD NETWORK CONTEXT IS OPTIMAL FOR TRAJECTORY RECOVERY?

To determine the optimal amount of context, we need to balance information completeness with computational efficiency. While providing comprehensive road network data might seem beneficial, excessive context can overwhelm LLMs and degrade performance and significantly increase computational costs. We conducted experiments on the development set with the following configurations (Appendix F includes example prompts for each configuration):

(1) No network: An end-to-end approach where LLMs are asked to generate the final coordinates to recover the missing segments.

(2) Raw network - Direct: We include the full road network in the bounding box surrounding the masked segment for direct coordinate generation (the naive solution described in Section 4).

(3) Raw network - Two-stage: We use the two-stage approach described in Section 4, with the full road network in the bounding box surrounding the masked segment as above.

(4) Adjacent list - Two-stage: Similar to (3), while to improve readability, we transform the retrieved road network into an adjacency list-based graph representation, where each road includes explicit connection information, along with relevant road metadata such as road types, oneway constraint.

(5) Topology-only - Two-stage: Similar to (4), and we remove all road geometry, retaining only topology information.

(6) Topology-only + Direction - Two-stage (current system): Similar to (5), and we pre-compute the direction for each road segment in the road network by calculating its bearing using the road geometry to provide models with cues about the direction of the road, helping models have a general sense of the roads to choose to move toward the destination.

The results in Table 3 demonstrate a clear pattern where structured representations and explicit guidance significantly improve path finding performance. While the raw road network approaches struggled with information overload, using topology-only with direction guidance produces optimal

| Variant | PoT$_{F_1}$ ↑ | MAE$_{F_1}$ ↓ | Avg. # Tokens |
|---|---|---|---|
| No network | 39.9 | 13.0 | 780 |
| Raw network - Direct | 42.3 | 11.6 | 23 655 |
| Raw network - Two-stage | 45.2 | 11.3 | 25 947 |
| Adjacent list - Two-stage | 53.1 | 6.6 | 19 920 |
| Topology-only - Two-stage | 52.3 | 7.5 | 12 892 |
| Topology-only + Direction - Two-stage | **58.7** | **4.2** | 10 708 |

Table 3: Ablation result of GPT-4.1 on the GLOBALTRACE development set. Best scores are in **bold**, second-best are underlined.

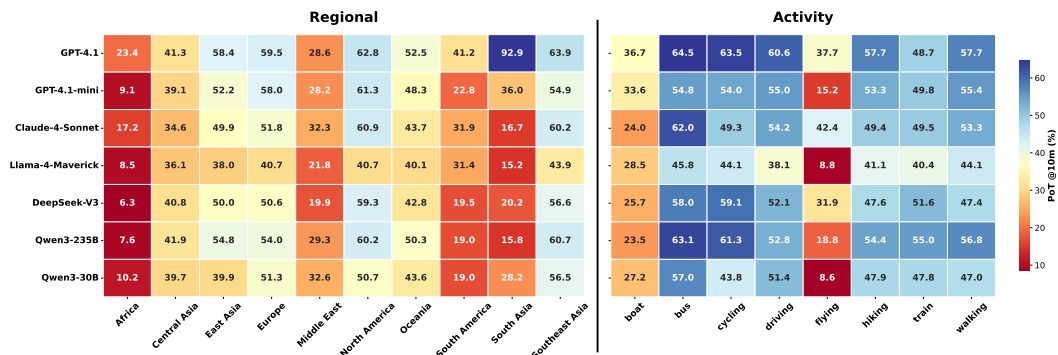

Figure 4: Reconstruction performance (PoT$_{F_1}$) of all LLMs tested, breakdown by regions and activity types.

performance by providing just the essential information LLMs need for effective path finding. The minimal performance difference between Adjacent list and Topology-only approaches suggests that geometric details beyond intersection points contribute little to path finding accuracy while substantially increasing computational overhead.

### 6.3 RQ3: Do LLMs Exhibit Geospatial Bias?

To further investigate potential biases, we breakdown the results based on activity types, regions, and training data cutoff date of LLMs.

**Regional Bias** From Figure 4, we can clearly see that the LLMs consistently perform worse on the Global South (Africa, Middle East, Oceania, South America, and South Asia) than they do on the Global North (Europe, North America, and Southeast Asia). The discrepancy between the two is non-trivial (with up to 50% difference, e.g., Africa vs. North America for Llama, DeepSeek, and Qwen). These results are unsurprising as most training data are concentrated to the western world, and LLMs would likely have more geospatial knowledge about those regions. One outlier is the abnormally high result of GPT-4.1 on South Asia. Most of those are hiking trails in Bhutan, which could potentially be a bias specific to this model. This finding also provides more context to a previous finding that LLMs are biased toward higher socioeconomic regions (Manvi et al., 2024a).

**Activity Type Bias** Cycling, bus, and driving are among the the best-performing activity types. These structured transportation modes utilize well-defined infrastructure. For these modes, LLMs consistently perform better. For pedestrian activities, i.e., walking and hiking, LLMs show poorer performance, while trajectories of flying and boat activities have the worst reconstruction quality. This pattern is consistent across all LLM models, suggesting systematic biases rather than model-specific limitations. The performance hierarchy appears to correlate with infrastructure definition and predictability: activities following established routes (roads, railways, and bike paths) achieve better reconstruction than free-form movements (e.g., walking and hiking).

**Data Contamination** We study the performance trend on traces before vs. after cutoff date for each model and observe no significant difference between the two periods, showing that LLMs do possess geospatial reasoning capabilities beyond just memorization (detailed in Appendix J).

### 6.4 RQ4: CAN LLMS INCORPORATE USER PREFERENCE TO THE NAVIGATION?

The finding that LLMs can comprehend road networks to construct paths offers new opportunities to enhance navigation experience beyond just finding the shortest or fastest routes between two locations. We conduct a case study to explore such opportunities.

In practice, when in familiar areas, people usually choose paths that suit their personal preferences, e.g., safe, scenic, close to water, or going through shops and cafes. There are several works that use POIs (e.g., landmarks, parks, and shops), environmental features (e.g., greenery, lighting, and street type) to plan travel (Ju et al., 2025; Wang et al., 2025; Yu et al., 2025a). The survey by Siriaraya et al. (2020) introduces the SWEEP (Safety, Well-being, Exploration, Effort, Pleasure) taxonomy, outlining different qualities in pedestrian paths. Based on SWEEP, we craft different scenarios, focusing mostly on the Exploration and Pleasure categories, embedding users' preferences and the relevant POIs to the context provided to the model. Below, we demonstrate one specific manually-crafted scenario. Refer to the full prompts and other scenarios in Appendix K.

**Urban Foodie** We select a scenario in Melbourne, famous for its diverse food scene. The start and end points are in two different ends of the city center, and the preference is a pedestrian path with many food options. The path generated by Google Maps is just a straightforward path that goes through the main streets with minimal detours. On the other hand, our system suggests a path that goes through alleys and lanes, where there could be more hidden gems and local options. Interestingly, the path managed to go through Hardware Lane, a famous alley for foods and drinks, despite this street was not retrieved as a POI. This show that the model can leverage its internal knowledge into route planning.

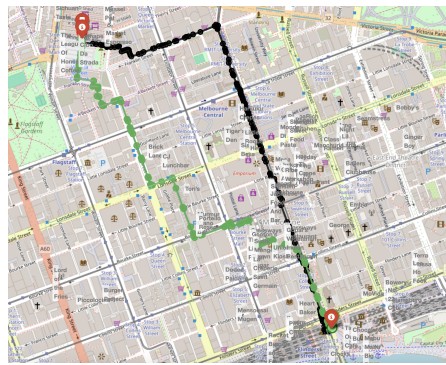

Figure 5: Comparison of "Urban foodie" paths generated by our system (green) and Google Maps (black)

**Additional Results** We also crafted the "First-time tourist" to show that our system works well in cases where Google Maps failed to suggest due to the lack of maps data, and "Waterfront cyclist" where model was able to suggest an alternative route that goes through more scenic POIs (Appendix K).

## 7 CONCLUSION

We introduced GLOBALTRACE, a benchmark for probing whether general-purpose LLMs can read road networks and reconstruct long masked trajectory segments without specialized training or external routing engines. Using a two-stage, schema-constrained framework, we showed that strong LLMs can produce valid, connected routes, outperforming specialized trajectory models and approaching the performance of an enterprise-grade navigation system. Our analysis suggests that structured map topology is key to elicit these capabilities, and reveals persistent gaps and biases across regions and transportation modes. Crucially, we also demonstrate that LLMs can go beyond shortest paths and flexibly integrate user preferences into route recommendation. In a broader sense, these findings show that LLMs have huge potential to support applications in mobility, accessibility, and urban decision making.

While our results demonstrate that LLMs possess non-trivial geospatial reasoning capabilities, several limitations remain. First, trajectory recovery, while informative, captures only a narrow slice of geospatial understanding. Second, the definition of "ground truth" recoveries is itself imperfect. Our benchmark is constructed from user-generated GPS traces, which may be noisy and reflect individual rather than standard choices of movement.

**Ethics Statement**    This work involves the use of user-generated data, in the form of GPS traces. Those data are uploaded voluntarily and contain no personal identifiable information (PII). One more concern is on sending those data to proprietary LLM APIs. Again, as those traces are readily-available and allow crawling through the official API, there should not be any violation regarding the terms and agreements of OpenStreetMap.

**Reproducibility Statement**    To ensure full reproducibility and to facilitate future research, we release the full GLOBALTRACE dataset (Section 3) along with the code to replicate the findings in this work (Section 6) at `https://anonymous.4open.science/r/llm_traj_rec-5F5B/`. We also document the settings that were used for each model (Appendix E).

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

## A  FULL DISCUSSION ON RELATED WORKS

**Trajectory recovery**   Traditional trajectory recovery methods have typically relied on probabilistic modeling and map-matching techniques to reconstruct missing points from noisy or sparse GPS traces. More recent, there are efforts that construct foundation models for trajectories. Most notably, TrajBERT (Si et al., 2024) introduces a BERT-style Transformer for sparse trajectories where location points are missing from poor GPS signal; it learns bidirectional temporal patterns with spatial-temporal refinement to recover missing points. Following the same direction, TrajFM (Lin et al., 2024) builds a vehicle trajectory foundation model for region and task transferability. The paper propose to integrate spatial, temporal, and POIs, and uses trajectory masking/recovery to pre-train a standard Transformer model. Instead of relying purely on coordinates-level data, BigCity (Yu et al., 2025b) merge trajectories with population-level traffic state data (e.g. flow, speed, density across road segments) to perform additional tasks like next-location prediction, and traffic forecasting. Most of these works, however, focus on point recoveries (rather than a large continuous segment). Furthermore, the biggest bottleneck is generalizability, as models trained for a specific regions are not transferrable to other unseen regions. To the best of our knowledge, our work is the first to explore this by applying general-purpose LLMs to this task, with the focus on zero-shot capabilities without the need of training or additional supervision signals.

**Geospatial understanding capabilities of LLMs**   Recent studies have probed whether large language models possess an implicit grasp of geographic space and spatial reasoning. Gurnee & Tegmark (2024) show that LLMs are are able to map coordinate to corresponding regions. Manvi et al. (2024b) show that LLMs embed substantial geospatial knowledge (e.g., population density and mean income), but simply feeding geographic coordinates into prompts yields poor results on those questions. The authors augment coordinate inputs with auxiliary map data (from OpenStreetMap), such as point-of-interest (POIs) around the coordinates, which lead to large improvement over naive baselines. Manvi et al. (2024a) further extend this analysis, showing that LLMs exhibit systematic geographic biases (e.g., against lower-socioeconomic status regions). MapEval (Dihan et al., 2025) is a benchmark of 700 multiple-choice, map-grounded questions (textual/API/visual) across 180 cities and 54 countries. Across 30 foundation models tested, none exceeds 67% accuracy and all lag behind human performance, with particular difficulties in distance/direction inference, route planning, and visual map understanding. Unlike previous task formulations of knowledge extraction, bias diagnostic, and multiple-choice question answering, we tackle trajectory recovery as long-segment route reconstruction and investigate whether general-purpose LLMs can perform this in a zero-shot manner, without in-domain pretraining or extra supervision.

## B  MORE DETAILS ON GLOBALTRACE

We show the cities, activity types, and regions covered by GLOBALTRACE in Table 4 to Table 6.

| Region | Cities |
| --- | --- |
| East Asia | Hong Kong, Mongolia, Seoul, Shanghai, Tokyo |
| Southeast Asia | Bangkok, Cambodia, Kuala Lumpur, Singapore, Vietnam |
| South Asia | Bhutan, Delhi, Nepal, Sri Lanka |
| Central Asia | Uzbekistan (Silk Road) |
| Middle East | Iran (Persepolis), Jordan (Petra) |
| Europe | Alps (Chamonix, Zermatt), Amsterdam, Aosta Valley, Athens, Barcelona, Bavarian Alps, Berlin, Black Forest, Bohemian Switzerland, Budapest, Carpathians, Copenhagen, Corsica, Dolomites, Durmitor, Helsinki, Istanbul, Jotunheimen, Julian Alps, Lake District, Lisbon, London, Madrid, Munich, Oslo, Paris, Plitvice Lakes, Prague, Pyrenees, Rome, Scottish Highlands, Scottish Isles, Stockholm, Swiss Jura, Tatra Mountains, Vienna, Vosges Mountains, Warsaw |
| North America | Montreal, New York, San Francisco, Toronto, Vancouver |
| South America | Bolivia, Buenos Aires, Iguazu Falls, Peru (Sacred Valley), Rio de Janeiro |
| Oceania | Chile (Easter Island), Fiji Islands, Melbourne, Sydney |
| Africa | Kenya, Tanzania (Serengeti) |

Table 4: Cities covered by GLOBALTRACE.

| City | Dev | Test | Train | boat | bus | cycling | driving | flying | hiking | train | walking | Total |
|---|---|---|---|---|---|---|---|---|---|---|---|---|
| alps_chamonix | 0 | 0 | 11 | 0 | 0 | 0 | 2 | 0 | 6 | 0 | 3 | 11 |
| alps_zermatt | 0 | 0 | 18 | 0 | 0 | 0 | 0 | 0 | 18 | 0 | 0 | 18 |
| amsterdam | 0 | 0 | 52 | 0 | 0 | 0 | 36 | 0 | 0 | 0 | 16 | 52 |
| aosta_valley | 0 | 0 | 80 | 0 | 0 | 39 | 3 | 0 | 35 | 0 | 3 | 80 |
| athens | 0 | 0 | 39 | 0 | 2 | 0 | 37 | 0 | 0 | 0 | 0 | 39 |
| bangkok | 0 | 4 | 0 | 2 | 1 | 0 | 0 | 1 | 0 | 0 | 0 | 4 |
| barcelona | 0 | 0 | 110 | 0 | 1 | 21 | 4 | 0 | 62 | 12 | 10 | 110 |
| bavarian_alps | 0 | 0 | 123 | 0 | 0 | 111 | 0 | 0 | 11 | 0 | 1 | 123 |
| berlin | 0 | 0 | 29 | 0 | 7 | 10 | 0 | 0 | 1 | 1 | 10 | 29 |
| bhutan | 0 | 3 | 0 | 0 | 0 | 0 | 0 | 0 | 3 | 0 | 0 | 3 |
| black_forest | 0 | 0 | 127 | 0 | 0 | 7 | 4 | 0 | 75 | 21 | 20 | 127 |
| bohemian_switzerland | 0 | 0 | 75 | 0 | 0 | 1 | 1 | 1 | 71 | 0 | 1 | 75 |
| bolivia | 0 | 0 | 98 | 0 | 0 | 5 | 2 | 0 | 85 | 0 | 6 | 98 |
| budapest | 0 | 0 | 21 | 13 | 0 | 1 | 0 | 0 | 2 | 2 | 3 | 21 |
| buenos_aires | 0 | 1 | 0 | 0 | 0 | 0 | 0 | 0 | 0 | 0 | 1 | 1 |
| cambodia | 0 | 1 | 0 | 0 | 0 | 1 | 0 | 0 | 0 | 0 | 0 | 1 |
| carpathians | 0 | 0 | 321 | 2 | 4 | 21 | 29 | 0 | 196 | 24 | 45 | 321 |
| chile_easter_island | 37 | 0 | 0 | 1 | 0 | 9 | 0 | 0 | 25 | 0 | 2 | 37 |
| copenhagen | 0 | 0 | 84 | 0 | 0 | 0 | 0 | 0 | 0 | 0 | 84 | 84 |
| corsica_gr20 | 0 | 0 | 207 | 0 | 0 | 15 | 95 | 0 | 97 | 0 | 0 | 207 |
| delhi | 22 | 0 | 0 | 0 | 14 | 0 | 1 | 0 | 0 | 6 | 1 | 22 |
| dolomites | 0 | 0 | 8 | 0 | 0 | 0 | 0 | 0 | 8 | 0 | 0 | 8 |
| durmitor | 0 | 0 | 8 | 0 | 0 | 0 | 0 | 0 | 8 | 0 | 0 | 8 |
| fiji_islands | 0 | 12 | 0 | 0 | 0 | 0 | 4 | 0 | 7 | 0 | 1 | 12 |
| helsinki | 2 | 0 | 0 | 0 | 0 | 0 | 0 | 0 | 0 | 0 | 2 | 2 |
| hong_kong | 0 | 32 | 0 | 0 | 0 | 0 | 16 | 2 | 8 | 6 | 0 | 32 |
| iguazu_falls | 0 | 0 | 12 | 0 | 0 | 0 | 0 | 0 | 12 | 0 | 0 | 12 |
| iran_persepolis | 0 | 30 | 0 | 0 | 0 | 0 | 26 | 0 | 1 | 0 | 3 | 30 |
| istanbul | 0 | 49 | 0 | 0 | 0 | 0 | 49 | 0 | 0 | 0 | 0 | 49 |
| jordan_petra | 88 | 0 | 0 | 0 | 0 | 85 | 3 | 0 | 0 | 0 | 0 | 88 |
| jotunheimen | 0 | 0 | 15 | 0 | 0 | 15 | 0 | 0 | 0 | 0 | 0 | 15 |
| julian_alps | 0 | 0 | 12 | 0 | 0 | 3 | 2 | 0 | 6 | 0 | 1 | 12 |
| kenya | 0 | 1 | 0 | 0 | 0 | 0 | 1 | 0 | 0 | 0 | 0 | 1 |
| kuala_lumpur | 0 | 198 | 0 | 0 | 85 | 1 | 76 | 0 | 3 | 24 | 9 | 198 |
| lake_district | 0 | 0 | 50 | 0 | 0 | 0 | 0 | 0 | 50 | 0 | 0 | 50 |
| lisbon | 0 | 0 | 16 | 0 | 0 | 0 | 0 | 0 | 5 | 0 | 11 | 16 |
| london | 0 | 0 | 54 | 0 | 2 | 8 | 3 | 0 | 2 | 2 | 37 | 54 |
| madrid | 0 | 0 | 21 | 0 | 6 | 0 | 0 | 0 | 2 | 0 | 13 | 21 |
| melbourne | 0 | 18 | 0 | 0 | 0 | 6 | 0 | 0 | 4 | 2 | 6 | 18 |
| mongolia | 0 | 9 | 0 | 0 | 0 | 1 | 4 | 0 | 3 | 0 | 1 | 9 |
| montreal | 0 | 0 | 213 | 0 | 1 | 10 | 101 | 0 | 0 | 6 | 95 | 213 |
| munich | 0 | 0 | 10 | 0 | 0 | 1 | 1 | 1 | 0 | 2 | 5 | 10 |
| nepal | 0 | 0 | 258 | 0 | 29 | 1 | 46 | 0 | 177 | 0 | 5 | 258 |
| new_york | 85 | 0 | 0 | 2 | 0 | 33 | 11 | 0 | 0 | 2 | 37 | 85 |
| oslo | 1 | 0 | 0 | 0 | 0 | 0 | 0 | 0 | 1 | 0 | 0 | 1 |
| paris | 0 | 0 | 17 | 0 | 6 | 1 | 1 | 0 | 0 | 2 | 7 | 17 |
| peru_sacred_valley | 0 | 0 | 107 | 28 | 0 | 1 | 25 | 1 | 36 | 0 | 16 | 107 |
| plitvice_lakes | 0 | 0 | 31 | 0 | 0 | 1 | 28 | 0 | 2 | 0 | 0 | 31 |
| prague | 0 | 0 | 3 | 0 | 0 | 0 | 0 | 0 | 2 | 0 | 1 | 3 |
| pyrenees | 0 | 0 | 14 | 0 | 0 | 0 | 0 | 0 | 14 | 0 | 0 | 14 |
| rio_de_janeiro | 0 | 2 | 0 | 0 | 0 | 0 | 0 | 0 | 2 | 0 | 0 | 2 |
| rome | 0 | 0 | 91 | 0 | 3 | 28 | 43 | 0 | 13 | 1 | 3 | 91 |
| san_francisco | 0 | 50 | 0 | 0 | 0 | 0 | 37 | 0 | 3 | 0 | 10 | 50 |
| scottish_highlands | 0 | 0 | 89 | 0 | 0 | 1 | 0 | 0 | 88 | 0 | 0 | 89 |
| scottish_isles | 0 | 0 | 8 | 0 | 0 | 0 | 0 | 0 | 8 | 0 | 0 | 8 |
| seoul | 0 | 0 | 97 | 19 | 4 | 39 | 0 | 0 | 5 | 5 | 25 | 97 |
| shanghai | 60 | 0 | 60 | 0 | 24 | 0 | 12 | 4 | 28 | 10 | 42 | 120 |
| singapore | 16 | 0 | 0 | 0 | 12 | 0 | 3 | 0 | 1 | 0 | 0 | 16 |
| sri_lanka | 0 | 0 | 2 | 0 | 0 | 0 | 1 | 0 | 1 | 0 | 0 | 2 |
| stockholm | 0 | 0 | 37 | 12 | 0 | 1 | 7 | 0 | 6 | 0 | 11 | 37 |
| swiss_jura | 0 | 0 | 38 | 0 | 0 | 35 | 0 | 0 | 2 | 0 | 1 | 38 |
| sydney | 0 | 39 | 0 | 0 | 0 | 0 | 0 | 0 | 39 | 0 | 0 | 39 |
| tanzania_serengeti | 0 | 1 | 0 | 0 | 0 | 0 | 1 | 0 | 0 | 0 | 0 | 1 |
| tatra_mountains | 0 | 0 | 40 | 0 | 0 | 1 | 0 | 0 | 32 | 6 | 1 | 40 |
| tokyo | 0 | 0 | 187 | 0 | 0 | 2 | 13 | 3 | 9 | 66 | 94 | 187 |
| toronto | 0 | 3 | 0 | 0 | 0 | 0 | 3 | 0 | 0 | 0 | 0 | 3 |
| uzbekistan_silk_road | 0 | 17 | 0 | 0 | 0 | 0 | 14 | 0 | 3 | 0 | 0 | 17 |
| vancouver | 41 | 0 | 0 | 0 | 30 | 0 | 2 | 0 | 9 | 0 | 0 | 41 |
| vienna | 0 | 99 | 0 | 0 | 1 | 3 | 0 | 0 | 75 | 3 | 17 | 99 |
| vietnam | 244 | 0 | 0 | 1 | 0 | 0 | 202 | 2 | 13 | 16 | 10 | 244 |
| vosges_mountains | 0 | 0 | 10 | 0 | 0 | 0 | 0 | 0 | 9 | 0 | 1 | 10 |
| warsaw | 0 | 0 | 27 | 0 | 0 | 1 | 1 | 0 | 18 | 1 | 6 | 27 |
| **Total** | 596 | 569 | 2930 | 80 | 232 | 519 | 950 | 15 | 1402 | 220 | 677 | 4095 |

Table 5: GLOBALTRACE dataset statistics by region and activity type.

| Geographical Region | Regions | Dev | Test | Train | Total Traces |
|---|---|---|---|---|---|
| Europe | 39 | 3 | 148 | 1896 | 2047 |
| East Asia | 5 | 60 | 41 | 344 | 445 |
| Southeast Asia | 5 | 260 | 203 | 0 | 463 |
| South Asia | 4 | 22 | 3 | 260 | 285 |
| Central Asia | 1 | 0 | 17 | 0 | 17 |
| Middle East | 2 | 88 | 30 | 0 | 118 |
| North America | 5 | 126 | 53 | 213 | 392 |
| South America | 5 | 0 | 3 | 217 | 220 |
| Africa | 2 | 0 | 2 | 0 | 2 |
| Oceania | 4 | 37 | 69 | 0 | 106 |
| **Total** | **72** | **596** | **569** | **2930** | **4095** |

Table 6: GLOBALTRACE dataset coverage by geographical regions.

## C  MORE DETAILS IN ROAD NETWORK CONSTRUCTION

We retrieve a road network surrounding the masked segment using the Overpass API (Overpass API, 2025). Given the start and end coordinates for the masked segment, we draw a rectangle around the straight line between the start and end points. To ensure enough coverage, we also expand this rectangle on all sides using a gap-aware buffer in meters (150 m for small gap and 500 m for large gap). This yields a compact box that encloses the segment rather than a large city-wide area. Inside this bounding box, we then retrieve relevant roads specific to the type of activity:

- Walking/Hiking: footway, pedestrian, path, steps, living_street, track, bridleway, road, residential, service, unclassified, tertiary, tertiary_link, secondary, secondary_link, primary, primary_link, cycleway, trunk, trunk_link

- Cycling: cycleway, path, living_street, track, residential, service, unclassified, tertiary, tertiary_link, secondary, secondary_link, primary, primary_link

- Driving/Bus: motorway, motorway_link, trunk, trunk_link, primary, primary_link, secondary, secondary_link, tertiary, tertiary_link, unclassified, residential, service

- Train: public_transport=station, railway=station, railway=subway_entrance, public_transport=platform/stop_position.

- Flying/Boat: No specific filtering

An example of a query for walking activity would look like:

Endpoint: `https://overpass-api.de/api/interpreter`

Request:

```
[out:json][timeout:30];
(
  way
    [highway~"footway|pedestrian|path|steps|living_street|track|bridleway|
             road|residential|service|unclassified|tertiary|tertiary_link|
             secondary|secondary_link|primary|primary_link|cycleway"]
      (48.2080,16.3600,48.2160,16.3800);
);
out geom;
```

## D  SAMPLE PROMPTS AND RESPONSES

**Path Selection**

**Prompt**

You are a navigation expert. Create a connected path from start to end point.
Start: [-37.6015900, 145.0244730]
End: [-37.6024230, 145.0274570]
Activity: CYCLING
Distance: 279m

— CONTEXT BEFORE —
- Starting near Equestria Boulevard, continuing along Grandidier Street
- The traveler was heading east before entering the masked segment
- Average speed: 8.3 m/s (steady cycling)

— CONTEXT AFTER —
Route narrative: Traveler continues east and will connect to Landover Road.
Movement: Expected speed steady at ∼8 m/s.
Behavioral interpretation: Still cycling straight, no significant detours.

— START POINT ANALYSIS —
Snapped to: unnamed cycleway (id=1347174722), confidence = 0.98
Bearing at entry: ∼95.7° (eastward)
Next candidate node: (node_id_original=12461706443)

— END POINT ANALYSIS —
Snapped to: Landover Road (id=1155201336), confidence = 1.00
Required approach bearing: ∼180°
Nearest junction: (node_id_original=12461663891)

— ROAD NETWORK (excerpt; verbatim) —

```
{"roads": {
  "1347174722": {"type":"cycleway","connects_to":[
    {"road_id":"1347175623","intersection_id":"12461706443",
     "coords":[-37.601601,145.024567],"bearing_to_dest":109.7}],
    "direction":"E"},
  "1347175623": {"type":"cycleway","connects_to":[
    {"road_id":"1347174722","intersection_id":"12461706443",
     "coords":[-37.601601,145.024567],"bearing_to_dest":109.7},
    {"road_id":"1347176804","intersection_id":"12461729290",
     "coords":[-37.601513,145.024983],"bearing_to_dest":114.9},
    {"road_id":"1377168420","intersection_id":"12461729290",
     "coords":[-37.601513,145.024983],"bearing_to_dest":114.9}],
    "direction":"E"},
    ...
}}
```

— TASK —
Choose a logical path from start to end point based on the activity and distance.
Output step-by-step navigation with road names, IDs, and intersections.

— EVALUATION REQUIREMENTS —
1. Path must be physically connected via shared intersections.
2. Max 3 steps (distance = 279m).
3. Each step must include:
• Direction (e.g., east, southeast). • Road name + ID (e.g., `cycleway (id=1347175623)`).
• Target intersection ID (`node_id_original=XXXX`).
4. Prefer "continue straight" over turns.
5. Do not include coordinate lists in step descriptions.

— OUTPUT FORMAT —
REASONING: Justification for path choice.

STEP-BY-STEP NAVIGATION: Structured steps following schema.

**Response**

```
**REASONING:**
The start point is on cycleway (id=1347174722), heading east,
consistent with pre-segment direction (~95).
The direct continuation is via cycleway (id=1347175623),
then southeast via cycleway (id=1347176650),
leading to the endpoint on Landover Road (id=1155201336).
This minimizes turns and keeps within cycleway infrastructure.

**STEP-BY-STEP NAVIGATION:**

step_1: From start point, travel east along cycleway (id=1347174722)
        until reaching intersection with cycleway (id=1347175623)
        (node_id_original=12461706443)

step_2: Continue straight east onto cycleway (id=1347175623)
        until intersection with cycleway (id=1347176650)
        (node_id_original=12461734106)

step_3: Continue southeast onto cycleway (id=1347176650)
        until reaching endpoint on Landover Road (id=1155201336)
        (node_id_original=12461663891)

**VALIDATION:**
- Step count = 3 (within 3-step limit).
- Each step includes direction, road name + ID, and intersection
    anchor.
- All roads are connected via listed node IDs.
- No coordinates included in step descriptions.
- Endpoint matches required destination on Landover Road.
```

### Coordinate Generation

**Prompt**

**TASK:** Generate coordinates for step_2 from the geometry below.
**STEP_2 DESCRIPTION:** step_2: Continue straight east onto connecting cycleway (id=1347175623) until intersection with cycleway (id=1347176650) (node_id_original=12461729290)
**GEOMETRY (excerpt):**

```
{
 "roads": [
   {
    "id": 1347175623,
    "name": "unnamed",
    "geometry": [
      [-37.6016014, 145.0245667],
      [-37.6016227, 145.0247631],
      [-37.6015604, 145.0248218],
      [-37.6015206, 145.0248792],
      [-37.6015129, 145.0249834]
    ]
   },
   {
    "id": 1347176650,
    "name": "unnamed",
```

```
    "geometry": [
      [-37.6015257, 145.0250507],
      [-37.6015590, 145.0251463],
      [-37.6015986, 145.0251966],
      [-37.6016346, 145.0252310],
      [-37.6016781, 145.0252713]
    ]
  }
 ],
 "intersections": [
   {"id": 12461729290, "lat": -37.6015000, "lon": 145.0250000}
 ]
}
```

Starting coordinate: [-37.6016000, 145.0246000] (from step_1)
**GENERATE ONLY THE CONTINUATION OF THIS LIST, STARTING WITH A COMMA:**

**Response**

```
step_2: Continue straight east onto connecting cycleway (id
    =1347175623)
until intersection with cycleway (id=1347176650) (node_id_original
    =12461729290)
- [[-37.6016000, 145.0246000],
  [-37.6016014, 145.0245667],
  [-37.6016227, 145.0247631],
  [-37.6015604, 145.0248218],
  [-37.6015206, 145.0248792],
  [-37.6015129, 145.0249834],
  [-37.6015000, 145.0250000]]
```

## E  MODEL DETAILS

**Linear**    Reconstruct the masked segment by connecting the segment's start and end GPS coordinates with a straight line and sampling evenly along that line.

**Linear + HMM**    Produce a simple, connected candidate polyline via linear interpolation, then snap it to the road/path network using a standard HMM-based map-matching service. For this step, we call the OSRM's HMM-based Map Matching API. The HMM combines (1) emission probabilities: likelihood of each observed point being near a road candidate (distance-based) and (2) transition probabilities: likelihood of moving between candidates given road graph connectivity and plausible travel costs; OSRM encapsulates this logic and returns a most-likely path (Viterbi).

**TrajFM**    We adapt the pre-trained TrajFM to GlobalTrace as a masked coordinate recovery model by aligning data, projection, and supervision to the task. Concretely, we keep the TrajFM architecture fixed and load pre-trained weights, then project trajectories into metric space using per-group UTM coordinates with the correct zone. We convert GlobalTrace supervision into the model's tokenized format using a task-specific padder that masks only spatial features at indices flagged by the dataset's mask column, and finetune with the following settings:

**LLMs**

| Hyperparameter | Value |
|---|---|
| Optimizer | Adam (weight decay $1 \times 10^{-6}$) |
| Learning rate | $5 \times 10^{-5}$ |
| Epochs | 20 |
| Scheduler | CosineAnnealingLR ($T_{\max}$=20, $\eta_{\min}$=$1 \times 10^{-6}$) |
| Batch size | 16 |
| Loss weights | spatial=2.0, temporal=0.5, token=0.5 |
| Gradient clipping | max_norm=1.0 |
| Task/Padder | gt_mask (masked coordinate recovery) |

Table 7: Finetuning hyperparameters for TrajFM

| Model | Exact Checkpoint | Param Size | Training Cutoff |
|---|---|---|---|
| GPT-4.1 | gpt-4.1 | Not disclosed | 2024-06-01 |
| GPT-4.1-mini | gpt-4.1-mini | Not disclosed | 2024-06-01 |
| Claude Sonnet-4 | anthropic/claude-sonnet-4 | Not disclosed | 2025-03-01 |
| Meta-Llama 4 (Maverick) | meta-llama/llama-4-maverick | 400B (17B active) | 2024-08-01 |
| DeepSeek v3 | deepseek/deepseek-chat-v3 (incl. -0324 variant) | 671B (37B active) | 2024-07-01 |
| Qwen3-235B | qwen/qwen3-235b (variant: qwen/qwen3-235b-a22b-2507) | 235B | 2024-06-01 (Estimated) |
| Qwen3-30B | qwen/qwen3-30b (variant: qwen/qwen3-30b-a3b-instruct-2507) | 30B | 2024-06-01 (Estimated) |

Table 8: LLM settings used in experiments: checkpoint, parameter size (if known), and training cutoff date.

# F   MORE DETAILS ON ABLATION STUDY

## RAW ROAD NETWORK

**Context**

```
--- RAW ROAD NETWORK DATA ---
Raw Road Network Data (Full OSM JSON):

{
  "43981478": {
    "id": 43981478,
    "name": "Road 43981478",
    "type": "unknown",
    "geometry": [
      [
        1.3916361,
        103.5465912
      ],
      [
        1.3910125,
        103.5465948
      ],
      [
        1.3899776,
        103.5466008
      ],
      ... (truncated)
    ],
    "oneway": "no",
    "access": null,
    "surface": null,
    "lanes": null,
    "maxspeed": null,
    "bridge": null,
    "tunnel": null,
```

```
    "nodes_osmid": [
     5863015808,
     8524349683,
     7542016752,
     7044866205,
     6090239253,
     7044866215,
     7541977250,
     7042850022,
     1911020654
    ]
   },
   "180657476": {
    "id": 180657476,
    "name": "Road 180657476",
    "type": "unknown",
    "geometry": [
     [
      1.382216,
      103.5515759
     ],
     [
      1.3812584,
      103.5530091
     ]
     ... (truncated)
    ],
    "oneway": "no",
    "access": null,
    "surface": null,
    "lanes": null,
    "maxspeed": null,
    "bridge": null,
    "tunnel": null,
    "nodes_osmid": [
     1911020646,
     10574515989
    ]
   },
   "180657527": {
    "id": 180657527,
    "name": "Road 180657527",
    "type": "unknown",
    "geometry": [
     [
      1.3845596,
      103.5479925
     ],
     [
      1.382216,
      103.5515759
     ]
     ... (truncated)
    ],
    "oneway": "no",
    "access": null,
    "surface": null,
    "lanes": null,
    "maxspeed": null,
    "bridge": null,
    "tunnel": null,
    "nodes_osmid": [
     1911020652,
```

```
        1911020646
      ]
    },
    "180657531": {
      "id": 180657531,
      "name": "Road 180657531",
      "type": "unknown",
      "geometry": [
        [
          1.3853906,
          103.5467324
        ],
        [
          1.385056,
          103.5472046
        ]
        ... (truncated)
      ],
      "oneway": "no",
      "access": null,
      "surface": null,
      "lanes": null,
      "maxspeed": null,
      "bridge": null,
      "tunnel": null,
      "nodes_osmid": [
        1911020654,
        7042850021
      ]
    },
    "180657553": {
      "id": 180657553,
      "name": "Road 180657553",
      "type": "unknown",
      "geometry": [
        [
          1.3724152,
          103.5431087
        ],
        [
          1.3725283,
          103.5434894
        ],
        [
          1.3726949,
          103.5440499
        ],
        ... (truncated)
```

## ADJACENT LIST

**Context**

```
--- ROAD NETWORK (ADJACENCY LIST) ---
Road Network (Adjacency List with Full Geometry):

Road: Tonnelle Avenue (ID: 60430069, Type: trunk)
  Connects to:
    -> Road 1350138790 at intersection 11072081884 ([40.7704930,
       -74.0434580])
```

```
     -> Road 316621099 at intersection 3227567283 ([40.7715490,
        -74.0427250])
   Full Geometry (50 points): [40.7568214, -74.0543041] -> [40.7571288,
      -74.0540013] -> [40.7584890, -74.0525561] -> [... truncated]

 Road: Road 1350138790 (ID: 1350138790, Type: service)
   Connects to:
     -> Road 1192329114 at intersection 12489885821 ([40.7704090,
        -74.0432650])
     -> Road 1350138779 at intersection 12489885820 ([40.7710770,
        -74.0427570])
     -> Road 60430069 at intersection 11072081884 ([40.7704930,
        -74.0434580])
   Full Geometry (8 points): [40.7711525, -74.0429339] -> [40.7711244,
      -74.0428670] -> [40.7710768, -74.0427567] -> [... truncated]

 Road: Road 1350138779 (ID: 1350138779, Type: footway)
   Connects to:
     -> Road 1192760824 at intersection 11900950545 ([40.7712240,
        -74.0428280])
     -> Road 1350138780 at intersection 11900950524 ([40.7709810,
        -74.0428550])
     -> Road 1350138790 at intersection 12489885820 ([40.7710770,
        -74.0427570])
   Full Geometry (19 points): [40.7709807, -74.0428549] -> [40.7709897,
      -74.0428462] -> [40.7710349, -74.0427908] -> [... truncated]

 Road: Road 1350138780 (ID: 1350138780, Type: footway)
   Connects to:
     -> Road 1192329114 at intersection 11900950521 ([40.7706850,
        -74.0430820])
     -> Road 1350138779 at intersection 11900950524 ([40.7709810,
        -74.0428550])
   Full Geometry (4 points): [40.7706848, -74.0430825] -> [40.7706918,
      -74.0430771] -> [40.7709734, -74.0428609] -> [... truncated]

 Road: Road 1181351412 (ID: 1181351412, Type: proposed)
   Connects to:
     -> Road 1181351410 at intersection 10970996238 ([40.7708440,
        -74.0429640])
     -> Road 659183610 at intersection 10970996241 ([40.7704510,
        -74.0423890])
   Full Geometry (3 points): [40.7708444, -74.0429640] -> [40.7706320,
      -74.0426479] -> [40.7704509, -74.0423895]

...
```

## TOPOLOGY ONLY

**Context**

```
--- ROAD NETWORK (TOPOLOGY ONLY) ---
Road Network (Topology Only - No Geometry):

Road: Tonnelle Avenue (ID: 60430069, Type: trunk)
  Connects to:
    -> Road 1192329085 at intersection 11068635760
    -> Road 1350138790 at intersection 11072081884
    -> Road 371374395 at intersection 3749476730
    -> Road 371374396 at intersection 3749476733
```

```
Road: Tonnelle Avenue (ID: 316621100, Type: trunk)
  Connects to:
    -> Road 1192329085 at intersection 11068635761
    -> Road 1192329088 at intersection 11068635771

Road: Road 371374396 (ID: 371374396, Type: service)
  Connects to:
    -> Road 1192329113 at intersection 11068635986
    -> Road 371374397 at intersection 3749476735
    -> Road 371374471 at intersection 3749472151
    -> Road 60430069 at intersection 3749476733
    -> Road 895552436 at intersection 3749476731

Road: Road 1350138790 (ID: 1350138790, Type: service)
  Connects to:
    -> Road 1192329114 at intersection 12489885821
    -> Road 60430069 at intersection 11072081884

Road: Road 1181351413 (ID: 1181351413, Type: proposed)
  Connects to:
    -> Road 1181351411 at intersection 10970996240
    -> Road 1181351418 at intersection 10970996242

Road: Road 1181351412 (ID: 1181351412, Type: proposed)
  Connects to:
    -> Road 1181351410 at intersection 10970996238
    -> Road 659183610 at intersection 10970996241

...
```

## G   MORE DETAILS ON STAGE-BASED ANALYSIS METRICS

**Connectivity**   We first extract all mentions of road ID from the generated step-by-step navigation (in order). Then using this ordered list, we will cross-check each consecutive road ID pair against the provided road network (already transformed into topology-only format with explicit connection defined). Then the Connectivity is just the percentage of pairs that are connected out of the total number of pairs.

**Network adherence**   Also extract all mentions of road ID from the generated step-by-step naviga­tion. Using this list, we check against the provided road network to see if the IDs are actually exist. The score is just the percentage of valid road ID out of all the generated road IDs

**Geometry adherence**   Similar to road network adherence, but this metric compare whether the generate coordinates are actually presented in the provided road geometry. The score is then just the percentage of valid coordinates out of all the generated coordinates

**Bearing**   For each LLM-produced navigation step, we extract the intended cardinal direction (e.g., north, southeast) and map it to a canonical expected bearing in degrees: N=0, NE=45, E=90, SE=135, S=180, SW=225, W=270, NW=315. We then compute the actual bearing between the step's start_point and end_point coordinates and normalize it to [0, 360). The per-step bearing error is the circular angular distance between expected and actual: $error = min(|expected - actual|, 360 - |expected - actual|)$.

| Method | PoT F1 @10m (%) | MAE F1 (%) | PoT GT→REC (%) | PoT REC→GT (%) | MAE GT→REC (%) | MAE REC→GT (%) |
|---|---|---|---|---|---|---|
| GPT-4.1 | 63.3 | 5.4 | 79.0 | 59.3 | 5.1 | 25.9 |
| Claude-Sonnet-4 | 58.2 | 6.2 | 76.9 | 53.5 | 5.5 | 33.0 |
| GPT-4.1-mini | 57.0 | 5.9 | 73.1 | 54.0 | 5.7 | 28.3 |
| DeepSeek | 55.8 | 6.1 | 75.5 | 51.3 | 5.5 | 32.5 |
| Qwen-3-235B | 52.4 | 5.2 | 72.7 | 55.4 | 6.7 | 33.7 |
| Qwen-3-30B | 51.9 | 6.2 | 73.1 | 50.3 | 6.4 | 36.1 |
| Llama-4-Maverick | 48.7 | 5.6 | 80.7 | 40.9 | 4.3 | 40.9 |

Table 9: Combined metrics: F1 (symmetric) and directional (GT→REC, REC→GT) on test set.

| Method | PoT GT→REC (%) | PoT REC→GT (%) | MAE GT→REC (%) | MAE REC→GT (%) |
|---|---|---|---|---|
| *Large Language Models* | | | | |
| GPT-4.1 | 88.7 | 62.4 | 6.2 | 37.6 |
| Claude-Sonnet-4 | 87.1 | 56.4 | 6.7 | 47.1 |
| GPT-4.1-mini | 85.4 | 58.0 | 6.6 | 38.2 |
| DeepSeek | 87.3 | 55.0 | 6.3 | 44.8 |
| Qwen-3-235B | 83.0 | 58.0 | 8.1 | 48.3 |
| Qwen-3-30B | 86.1 | 53.8 | 7.4 | 50.4 |
| Llama-4-Maverick | 90.8 | 42.9 | 4.8 | 53.3 |

Table 10: Directional metrics: Small gap.

| Method | PoT GT→REC (%) | PoT REC→GT (%) | MAE GT→REC (%) | MAE REC→GT (%) |
|---|---|---|---|---|
| GPT-4.1 | 69.4 | 56.3 | 4.0 | 14.3 |
| Claude-Sonnet-4 | 66.8 | 50.6 | 4.3 | 19.0 |
| Qwen-3-235B | 62.1 | 52.6 | 5.3 | 18.6 |
| DeepSeek | 63.6 | 47.7 | 4.7 | 20.2 |
| GPT-4.1-mini | 60.9 | 50.0 | 4.7 | 18.4 |
| Llama-4-Maverick | 70.8 | 38.8 | 3.8 | 28.7 |
| Qwen-3-30B | 60.1 | 46.8 | 5.5 | 21.6 |

Table 11: Directional metrics: Large gap.

# H  DETAILED RESULTS

## H.1  RESULTS FOR EACH RECONSTRUCTION DIRECTION

A clear pattern shown in Table 9 to Table 11 is that many LLMs show high GT→REC and much lower REC→GT (e.g., good coverage of the ground-truth path but poor precision). Interpreting the directions: GT→REC measures recall (how much of the GT path is covered by the reconstruction), while REC→GT measures precision (how much of the reconstructed path lies on or near the GT). So a low REC→GT alongside a high GT→REC typically means the model covers most of the true route but also adds extra, off-route geometry—spurs, loops, or side streets—so a large share of reconstructed points are not close to the GT.

## H.2  ACTIVITY RESULTS

We show the activite bias results in Table 12.

| Model | boat | bus | cycling | driving | flying | hiking | train | walking |
|---|---|---|---|---|---|---|---|---|
| GPT-4.1 | 43.4 | 66.9 | 68.6 | 66.3 | 40.5 | 59.9 | 53.6 | 61.3 |
| GPT-4.1-mini | 43.4 | 57.3 | 57.4 | 59.7 | 21.4 | 54.0 | 54.4 | 57.5 |
| Claude-4-Sonnet | 31.6 | 64.7 | 55.4 | 60.8 | 47.6 | 52.5 | 55.5 | 56.7 |
| Llama-4-Maverick | 25.8 | 54.0 | 52.7 | 47.1 | 13.2 | 47.4 | 48.7 | 51.9 |
| DeepSeek-V3 | 35.1 | 61.0 | 61.1 | 57.6 | 37.0 | 51.6 | 56.5 | 51.1 |
| Qwen3-235B | 33.1 | 58.2 | 55.2 | 53.7 | 26.8 | 47.8 | 52.2 | 51.5 |
| Qwen3-30B | 36.5 | 56.1 | 47.3 | 55.3 | 13.9 | 47.4 | 49.9 | 48.6 |

Table 12: Activity bias results: mean PoT (%) by model and activity

## H.3 REGIONAL RESULTS

We show the regional bias results in Table 13.

| Model | Africa | Central Asia | East Asia | Europe | Middle East | North America | Oceania | South America | South Asia | Southeast Asia |
|---|---|---|---|---|---|---|---|---|---|---|
| GPT-4.1 | 33.1 | 50.2 | 62.0 | 65.2 | 35.7 | 67.4 | 52.5 | 47.4 | 81.1 | 67.4 |
| GPT-4.1-mini | 15.1 | 46.4 | 53.6 | 61.2 | 32.7 | 64.5 | 48.7 | 34.2 | 42.4 | 58.2 |
| Claude-4-Sonnet | 23.3 | 42.9 | 54.3 | 58.3 | 41.1 | 65.7 | 44.2 | 40.9 | 26.2 | 64.5 |
| Llama-4-Maverick | 14.2 | 46.4 | 45.3 | 50.1 | 29.9 | 48.8 | 44.2 | 39.9 | 24.6 | 51.7 |
| DeepSeek-V3 | 10.9 | 46.7 | 51.7 | 56.4 | 27.9 | 64.6 | 45.8 | 30.9 | 30.1 | 60.4 |
| Qwen3-235B | 12.7 | 38.4 | 49.8 | 51.3 | 31.2 | 61.8 | 42.6 | 30.4 | 26.2 | 57.7 |
| Qwen3-30B | 16.9 | 44.6 | 40.8 | 54.0 | 33.6 | 56.1 | 41.3 | 29.0 | 28.1 | 57.5 |

Table 13: Regional bias results: mean PoT (%) by model and region.

## I GOOGLE MAPS ERRORS

The mismatch between PoT and MAE also reveals some interesting insights. For linear and our two-stage LLM-based solutions, we notice that the MAE scores are strong, while Google Maps clearly favors PoT.[1] We notice that Google Maps often takes realistic detours to adhere to road rules, which may not match well for free-form movements like walking or hiking. Figure 6 shows an example. In Figure 6a, Google Maps is constrained to follow the road direction (the black line), achieving a lower MAE to the user trajectory (dark green), which takes a parallel segment that run in the other direction. This shows PoT is more robust to evaluate those cases as the two trajectories are within the same corridor. Also, in areas without sufficient road network data, Google Maps may generate a different trajectory (Figure 6b, or may not be able to generate any route at all (see Appendix K)

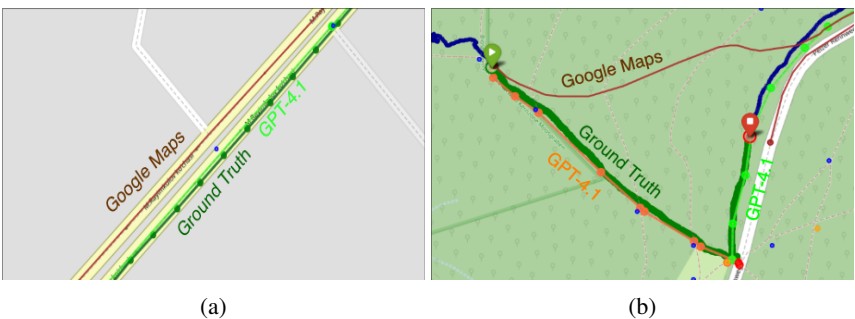

(a)                                                    (b)

Figure 6: Comparison of Google Maps (brown) vs. our system in walking activity. Our system (light green, orange) follows the ground truth (dark green) trajectories closely in both cases

## J CUTOFF ANALYSIS

LLMs training corpora contains a huge amount of geolocation data (Ilyankou et al., 2024), raising a concern of data leakage given their strong performance. We observe the performance trend before vs. cutoff date for each model. To facilitate a fair comparison, we perform stratification to get a balance distribution in term of on activity types, regions, and masking strategies for the pre- and post-cutoff sets. Overall, we observe no significant different between the two periods (Figure 7), where most models in fact have slightly better performance post cutoff date, showing that LLMs do posses geospatial reasoning capabilities rather than just memorization. Regardless, given that all the traces are from 2024 onward, it is unlikely that data contamination is a concern as it has been shown that the LLMs have a much more distant effective knowledge cutoff date compared to the reported date (Cheng et al., 2024).

---

[1] Also the low value of MAE can be misleading as it is normalized by the masked segment length. For example, a 5% error for a 2 km masked segment is 100 m, which is actually quite high, while a 15% error of a 200 m masked segment is just 30 m.

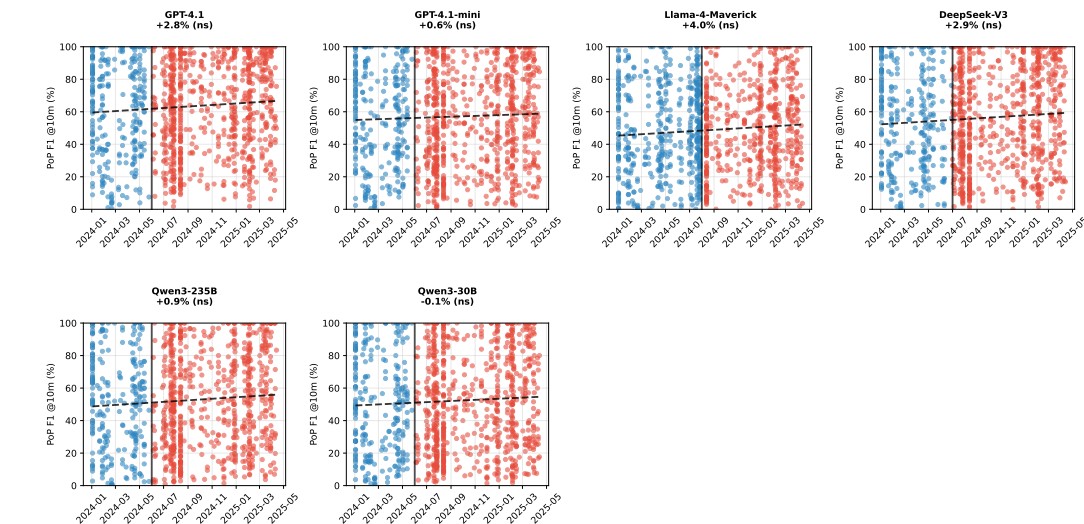

Figure 7: Reconstruction performance (PoT) of pre- vs. post-cutoff date for each model. **ns** denotes a non-significance change based on an independent t-test. Given a more recent cutoff date, Claude-4-Sonnet was excluded as we cannot find enough samples for post-cutoff analysis.

## K  PREFERENCE DEMO

> **Prompt**
>
> ```
> PREFERENCE-AWARE CONTEXT (for planning):
>
> USER PROFILE: Urban Culinary Corridor
> Description: Food/market/shopping corridor via pleasant pedestrian
>     streets
>
> ROUTING PRIORITIES (ordered):
> - proximity to food_and_drink, markets, shopping, pedestrian_areas
> - scenic beauty and interesting views
> - safety and pedestrian infrastructure
>
>
> ROUTE LENGTH + EFFORT CONSTRAINTS:
> - Direct distance: ~1713 m
> - Target total length: 1628-2485 m (hard max: 2485 m)
> - Maintain balance between exploration and effort: avoid unnecessary
>     detours, backtracking, or loops
> - Prefer corridor-aligned POIs and short deviations only when
>     warranted by preferences
> - Do not exceed 10 steps; typical is 3-7
>
> ANCHORING CONSTRAINTS:
> - step_1 MUST begin on a road within 60 m of the start coordinate.
>   * Prefer starting on: Unnamed road (id=721761316), distance=3m
> - The final step MUST end within 60 m of the destination.
>   * Prefer finishing on: Victoria Street (id=1153395320), distance=3m
> - Rules: Do not start step_1 on any road farther than 100 m from the
>     start.
>  Do not overshoot the destination; ensure the final coordinates end
>      exactly at the destination point.
>
> Start: [-37.8179000, 144.9691000]
> ```

```
End: [-37.8060000, 144.9567000]
Activity: WALKING

--- ROAD NETWORK ---
"17035879":{"id":17035879,"name":"McIntyre Alley","type":"service","
    connects_to":[{"road_id":17035877,"intersection_id":176693549}],"
    nearby_pois":[{"id":"593475843","name":"CrossCulture Church of
    Christ","category":"landmarks"},{"id":"2384426956","name":"The Big
     Clock","category":"landmarks"},{"id":"11867545682","name":"City
    on a Hill Melbourne","category":"landmarks"}]},"17035193":{"id
    ":17035193,"name":"Driver Lane","type":"service","connects_to":[{"
    road_id":291837736,"intersection_id":596909576},{"road_id
    ":715211398,"intersection_id":6722209690},{"road_id":715211389,"
    intersection_id":596909581}]},"22930862":{"id":22930862,"name":"
    Pender Place","type":"service","connects_to":[{"road_id
    ":715211379,"intersection_id":247018765},{"road_id":715211374,"
    intersection_id":6722197739}],"nearby_pois":[{"id":"2217921373","
    name":"Anglican Chinese Mission of the Epiphany","category":"
    landmarks"}]},
...
```

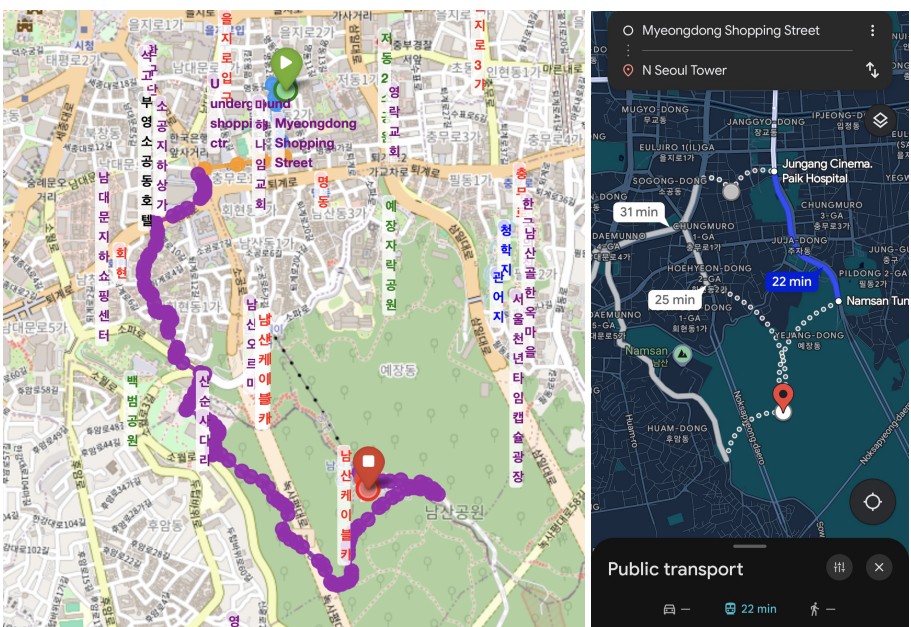

Figure 8: "First time tourist" routes comparison. Our systems route consists of steps with different colors, while Google Maps failed to suggest a valid pedestrian route.

**First-time Tourist**   Maximizes novel POIs coverage within distance/time; strings landmarks/markets/pedestrian areas; balances detours. For this scenario, we select a route between two popular tourist location in Seoul: From Myeongdong Shopping Street to N Seoul Tower. Google Maps was not able to suggest a walking route due to unavailable map data in this region while our system successfully construct a route going route various landmarks and popular tourist areas such as along Myeongdong Shopping Street, Underground shopping centre, Namsan Park.

**Scenic Cyclist**   Continuous rivers/lakes/bay boardwalks and promenades; avoids trunk roads. For this scenario, we select a trace in Sydney CBD, where there are a combination of waterfronts and gardens. Our system route start along the waterfront, similar to Google Maps, but then make a detour through several cycleways, and connect through the Royal Botanic Garden before moving

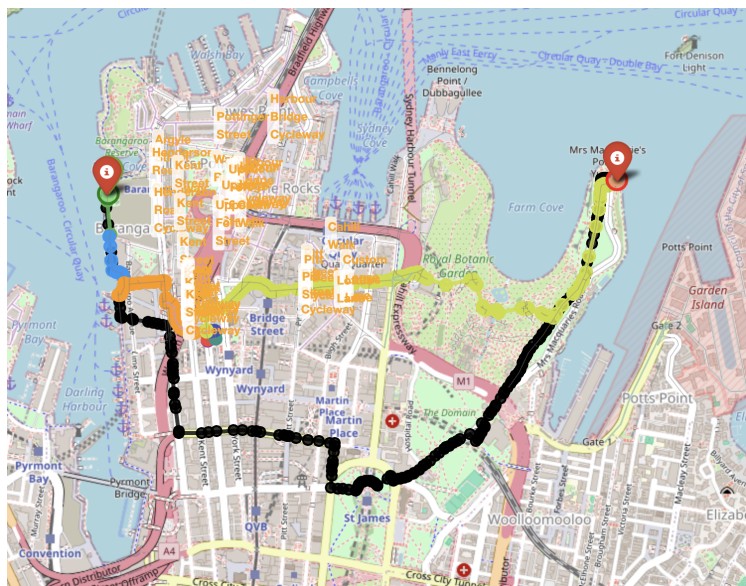

Figure 9: "Scenic cyclist" routes comparison. Our systems route consists of steps with different colors. The Google Maps route is in black.

toward the destination. Though this show that the generated route do follow user preference, our system output is not necessarily better compared to Google Maps.

## L    DISCLOSE OF LLM USAGE

We use GPT-5 through the web interface (https://chatgpt.com/) to aid in the writing of this paper, including generating latex commands to improve paper formatting and polish writing (grammar, collocation, rephrasing).

