# OpenReview forum: "Understanding the Geospatial Reasoning Capabilities of LLMs: A Trajectory Recovery Perspective"
_ICLR.cc/2026/Conference — ICLR 2026 Conference Withdrawn Submission_

### Official Review · Reviewer_Lcp1 · 2025-10-27

**Soundness:** 1
**Presentation:** 2
**Contribution:** 2
**Rating:** 4
**Confidence:** 4

**Summary:**

This paper investigates whether LLM can perform geospatial reasoning by reconstructing masked segments of GPS trajectories from road network context. The paper proposes a two-stage prompting framework, path selection followed by coordinate generation. Experiments show that LLMs outperform some baselines, generalize zero-shot across regions, and demonstrate structured reasoning over road networks. Further analyses reveal regional and activity-type biases and show that LLMs can integrate user preferences into route planning.

**Strengths:**

The introduced GLOBALTRACE benchmark is diverse and realistic.

The evaluation metrics go beyond standard MAE, providing more interpretable path-level measures such as PoT and PoTF1.

The paper is well-structured.

**Weaknesses:**

The method focuses on prompting design, but does not report any inference or token cost. Since some configurations involve 10K~20K tokens per query.

In Stage 1, the model receives a pre-selected local road network, which already restricts the possible routes to a small area, which may reduce task difficulty and partly explain its high accuracy compared to TrajFM.

Stage 2 relies on explicit road geometries, which introduces a strong spatial prior and reduces the need for actual geographic reasoning.

The current small/large gap split provides only a coarse difficulty granularity: the large-gap range is broad, and longer masked segments would more effectively evaluate genuine geospatial reasoning. In addition, the paper does not report how many samples fall under each difficulty level, which would be important for assessing the balance and relative contribution of each subset to the overall results.

**Questions:**

How much time does the two-stage prompting framework take to recover one trajectory in practice?

Does this strong geometric prior in both stages simplify the task (geospatial reasoning) too much?

In Fig. 2a, the average “Full trajectory (km)” differs between the small- and large-gap subsets, although each trajectory is said to have both masking variants. Could authors clarify these?

---

> ### Author Response · Authors · 2025-11-18
> **Response to Reviewer Lcp1 (1/2)**
>
> Thank you for your constructive feedbacks in helping improving the paper. We address the concerns below.
>
> **Token cost analysis**
>
> A: We provided token analysis in Table 3. From this one can easily derive the cost based on their choices of LLMs and the infrastructure used.
>
> **In Stage 1, the model receives a pre-selected local road network, which already restricts the possible routes to a small area, which may reduce task difficulty and partly explain its high accuracy compared to TrajFM.**
>
> A: Google Maps and Linear + HMM also require a road network. For TrajFM, while not having access to road network, this is a pre-trained model trained on a huge amount of trajectories data (and one would expect some level of generalizability such as predicting what point comes next to maintain a smooth continuity in movement). Furthermore, we also fine-tuned it to GLOBALTRACE. The reason why we include TrajFM is to compare to traditional (and strong) methods in trajectory recovery.
>
> Regarding the "small area", processing entire city networks would be computationally impossible and introduce noise. Our approach mirrors how humans navigate, focusing on relevant local context (why would one need to know the whole map of NYC if they just want to travel between some locations in Manhattan?)
>
> **Stage 2 relies on explicit road geometries, which introduces a strong spatial prior and reduces the need for actual geographic reasoning.**
>
> A:
>
> - Stage 1 tests genuine reasoning: Path selection requires understanding topology, connectivity, and movement patterns without coordinate information (only topology provided).
> - Stage 2 still requires spatial understanding: Models must:
>   + understand where the starting point is located on the road geometry,
>   + decide which coordinate to select next based on the textual cardinal direction, and
>   + generate points in the correct directional order.
>
>
> **The current small/large gap split provides only a coarse difficulty granularity...**
>
> A: Thanks for the suggestion. We ensure a balanced distribution between small and large gaps across all data splits (e.g., 569 small gaps and 538 large gaps in the test set; the discrepancy here is because some traces shorter than 500 meters cannot generate a large gap mask).
>
> Increasing the distance of the large gap would overload the context of the models (as the retrieved road network would grow too large). We acknowledge that this is a limitation of the current method but will only get better when LLMs become more capable and able to analyze larger input contexts.
>
> (cont.)

---

> ### Author Response · Authors · 2025-11-18
> **Response to Reviewer Lcp1 (2/2)**
>
> (cont.)
>
> **How much time does the two-stage prompting framework take to recover one trajectory in practice?**
>
> A: In our setup, using the GPT-4.1 model through OpenAI API, it takes around 5 seconds on average. Most of the time is spent on stage 2 where we provide the prompt for each generated navigation step iteratively.
>
> **Does this strong geometric prior in both stages simplify the task (geospatial reasoning) too much?**
>
> A: In Stage 1, the area of the retrieved road network is still very large as we expand the bounding box on all sides to ensure enough coverage of the area surrounding the trace, with each road accompanied by its own metadata. In Stage 2, models still need to reason using the provided geometries to decide which points to select. The far-from-perfect results of our method show that this is not a trivial task. Results of Google Maps are also far from perfect, showing that this task requires more than just large coverage.
>
> **In Fig. 2a, the average “Full trajectory (km)” differs between the small- and large-gap subsets, although each trajectory is said to have both masking variants. Could authors clarify these?**
>
> A: Thanks for pointing this out. Some trajectories with full length near the 500-meter minimum length cannot generate valid large gaps, hence skewing the distribution.

---

> ### Author Response · Authors · 2025-11-25
>
> Thank you for your time in engaging in the response period. As the rebuttal deadline is approaching, we would like to check if there are any remaining concerns or questions we could help clarify. We remain open to further discussion.

---

> > ### Comment · Reviewer_Lcp1 · 2025-11-25
> >
> > I thank the authors for the detailed responses. After reading the rebuttal and considering the views of the other reviewers, I believe my original assessment remains appropriate.
> >
> > First, although the newly introduced GLOBALTRACE benchmark is carefully constructed, it is ultimately a relatively small dataset of roughly 4,000 trajectories. This raises concerns about how representative or broadly generalizable the reported findings are, especially given the diversity of regions and activity types the paper aims to cover.
> >
> > Second, I agree with the other reviewers that the motivation and underlying intuition of the proposed two-stage prompting design are not yet sufficiently clear in the current version. While the paper presents extensive engineering effort, the core conceptual insight, the “why this works” rather than simply “how it is implemented”, remains hard to grasp from the reader’s perspective.
> >
> > Third, following the authors’ response regarding inference cost, I independently checked the approximate API expenses. For a single trajectory, GPT-4.1-mini appears to cost around $0.0043 (input tokens only), whereas GPT-4.1 can reach $0.02 per trajectory. In contrast, Google Maps, which delivers substantially higher performance on this task, costs about $0.005 per query. This raises a fundamental question: what is the practical advantage of using LLMs, especially given that the paper does not convincingly demonstrate the promised benefit of preference-aware navigation (as also noted by reviewers zLdE and 5qrW)?
> >
> > In summary, while the paper reflects substantial engineering effort, I remain unconvinced that the current method offers sufficient intuition, empirical insight, or practical justification to change my original evaluation.

---

> > > ### Author Response · Authors · 2025-12-02
> > >
> > > Thank you for your comment. We feel your concerns and want to clarify a few points.
> > >
> > > - We want to stress again that the main motivation for this work is on evaluating the geospatial reasoning capabilities of LLMs, and from that we explore broader impacts of the main finding that they can comprehend road network topology and understand GPS coordinate well (thus, the use case on preference-based navigation). The task of trajectory recovery is chosen as it facilitate a novel, and challenging task. To perform well, LLMs must demonstrate multiple different skills related to planning, spatial awareness, movement understanding, cardinal direction understanding. The two-stage framework is exactly how we want to answer the question of "why this works". By doing so, we could facilitate a fine-grained analysis on different qualities of the output.
> > >
> > > - We are not trying to make a point that using LLMs is better, or more cost effective than traditional navigation system, just that they are worth to explore due to their flexibility in understanding user queries (which Google Maps obviously doesn't offer).
> > >
> > > - While it is always ideal to have a larger dataset, we maintain the position that 4000+ traces are enough for a LLM benchmark and are sufficient for the experiments conducted in the paper.

---

### Official Review · Reviewer_5qrW · 2025-10-29

**Soundness:** 1
**Presentation:** 2
**Contribution:** 2
**Rating:** 2
**Confidence:** 4

**Summary:**

The authors attempt to show that LLMs can reconstruct trajectories to the same degree as traditional mapping platforms. This paper does too much though and the authors need to break the paper into different focus area.

**Strengths:**

Using trajectories to test whether LLMs can spatially reason is a good idea.

**Weaknesses:**

*The prompts were extremely detailed and there were no prompt sensitivity analyses. The authors want to claim that they showed the LLMs were on par with Google Maps but the inputs were very different and the detailed prompts all but ensured the systems would perfir reasonably.

* The research questions were introduced in the results section, and were never motivated.

*Not enough detail was provided on exactly what trajectories needed to be generated. Appendix B was too light on details.

* Results of different LLMs fell in the range of 50-60% in the best cases, this is barely better than flipping a coin so these results do no engender confidence.

*The authors claim to show their approach can incorporate user preferences, but just using a couple of examples is not sufficient, especially for their claims. And the authors fail to consider that humans do not consider trajectory reconstruction.The authors should focus on this study separately in another paper.

It was not clear why cycling, bus, and driving was included in the same study as pedestrian activities, such analyses necessarily require different map resolutions and none of this information was included. For the authors to make the claims they did, they need to include all information so people could replicate their studies.

**Questions:**

*Why would people want to use LLMs for path planning when deterministic algorithms do this quite well?

*Does the level of effort required in setting up the prompts to generate outputs that LLMs can handle justify the effort, especially in light of the success of traditional trajectory planning approaches?

*How brittle is this approach in light of slight prompt variations?

---

> ### Author Response · Authors · 2025-11-18
> **Response to Reviewer 5qrW (1/2)**
>
> Thank you for the constructive feedbacks. We address the concerns below.
>
> **Prompt sensitivity and fair comparison between methods**
>
> A:
> -  Section 6.2 and Table 3 present extensive ablation comparing 6 different prompting strategies, from minimal ("No network") to detailed ("Topology-only + Direction"). If the comment was on  sensitivity to linguistics variation (e.g., paraphrasing how we provide instructions), this is out of the scope of this paper. Moreover, this type of sensitivity should not be a problem for current LLMs. We also test with multiple LLMs, and the prompts work for them all so the prompts used are robust and not LLM-specific.
> - Google Maps API is not an LLM, and hence the input provided needs to be different. However, we ensure that  each baseline is provided the information needed to perform the task. Google Maps API would also use their internal maps data (as structured graph) in order to perform routing given the input start and end points. Our key finding is that LLMs can perform this task without specialized training or external routing engines.
>
>
> **Motivations of RQs not introduced**
>
> A: This is a presentation choice. We believe that the motivation for the research questions would come naturally based on  the gaps outlined in the Introduction:
>
> - RQ1: Can LLMs perform trajectory recovery? (Motivation: using trajectory recovery as a task to evaluate geospatial reasoning capabilities of LLMs)
> - RQ2: What context enables best performance? (Motivation: optimizing information presented to LLMs for the task)
> - RQ3: Do LLMs exhibit geospatial biases? (Motivation: a natural extension given the global coverage of our benchmark, also following findings from Manvi et al., 2024a)
> - RQ4: Can LLMs enable preference-aware navigation? (Motivation: exploring broader implication of the findings)
>
>
> **Results of different LLMs fell in the range of 50-60% in the best cases, this is barely better than flipping a coin so these results do no engender confidence.**
>
> A: There might be a confusion that this was a simple binary classification task. The chance performance for this task is not 50%. We are predicting a series of points (as GPS coordinates) and how those points deviate from the ground truth points (forming a connected path adhering to the road network). PoT is not binary success/failure. It measures the percentage of points within 10 meters of the correct trajectory. A PoT of 60% means that the model correctly reconstructs the majority of the path, not random performance.
>
>
>
>
> **The authors claim to show their approach can incorporate user preferences, but just using a couple of examples is not sufficient, especially for their claims. And the authors fail to consider that humans do not consider trajectory reconstruction.The authors should focus on this study separately in another paper.**
>
> A: We acknowledge the preference-aware navigation demonstration (Section 6.4) is preliminary. However:
>
> - It demonstrates feasibility and opens research directions
> - It also shows new capabilities beyond traditional systems
> - Integrating preference into navigation is also why we benchmark this task with LLMs - it allows extension to these preference aware navigation tasks that are difficult for traditional methods.
>
> **It was not clear why cycling, bus, and driving was included in the same study as pedestrian activities, such analyses necessarily require different map resolutions and none of this information was included. For the authors to make the claims they did, they need to include all information so people could replicate their studies.**
>
> A: The data source we use is road network (in the form of directed graphs with metadata) and contain all the information needed for all modes of transportations. Perhaps you misunderstood this with using maps in the form of images. We clearly state this in the Problem Statement, Appendix C about how we retrieve appropriate road types per activity, and some example prompts (e.g. Page 15 contains a road network slice with cycling road metadata).
>
> (cont.)

---

> ### Author Response · Authors · 2025-11-18
> **Response to Reviewer 5qrW (2/2)**
>
> (cont.)
>
> **Why would people want to use LLMs for path planning when deterministic algorithms do this quite well?**
>
> A: Traditional algorithms excel at shortest-path routing but cannot understand context, integrate preferences, and adjust the route flexibly. This is the reason why we demonstrate some use cases in RQ4 and more in Appendix K. Furthermore, as we are moving toward AGI, spatial reasoning is a core component of general intelligence. Our benchmark provides a valuable resource to evaluate this capability.
>
>
> **How brittle is this approach in light of slight prompt variations?**
>
> A: While we have not exhaustively tested prompt variations, our ablation study shows that:
> - Our core approach works across 7 different LLMs (Table 1)
> - Structured information presentation is the key to elicit LLMs’ trajectory recovery capabilities, not specific wording. (Table 3).
>
> If the comment was on  sensitivity to linguistics variation (e.g., paraphrasing the wording or languages in how we provide instruction), this is out of the scope of this paper. Moreover, this type of sensitivity should not be a major problem for current LLMs.

---

> > ### Comment · Reviewer_5qrW · 2025-11-20
> >
> > The authors are still trying to do too much and the bulk of my comments still apply. In addition, the authors want to claim there is no memorization going on for spatial bias, but it is not clear why any specific cutoff date would influence outcomes since this type of data is not as affected by recent events.
> >
> > If they want to claim that LLMs can go beyond shortest paths and flexibly integrate user preferences into route recommendation, they need to run a human-in-the-loop study. Their current approach to making such claims is unacceptable for a scientific paper.

---

> > > ### Author Response · Authors · 2025-11-21
> > > **Response to Reviewer 5qrW comment**
> > >
> > > Thank you for the quick response.
> > >
> > > - Regarding spatial bias, the idea of cutoff analysis is mainly to show verbatim memorization, that is whether the models memorize the exact same trace (matching every gps point with the exact sampling rate and floating points) and just recall from memory to do the reconstruction. Our analysis confirms this is not the case. It is true that models were trained on large amounts of spatial data, but the fact that they can solve novel problems (reconstructing unseen and newly constructed masked trajectories) implies some level of reasoning.
> > >
> > > - Regarding the preference-based navigation experiment, we want to stress that this is a use case conducted to demonstrate that models being able to read and reason over map data could have broader impact. We in no way state that this experiment can be used as evidence for a scientific claim. We will revise the writing to state this more explicitly.
> > >
> > > We hope this, along with our comments on other concerns could help make the paper clearer.

---

### Official Review · Reviewer_pXYj · 2025-10-31

**Soundness:** 3
**Presentation:** 3
**Contribution:** 2
**Rating:** 2
**Confidence:** 4

**Summary:**

The paper explores to understand the geospatial reasoning capabilities of large language models. They propose a benchmark named GlobalTrace consisting of 4000 real-world trajectories, LLMs are tasked with reproducing masked segments of a trajectory i.e a sequence of longitude and latitude points using a two-stage task description. The evaluation shows that existing llms exhibit a high degree of geospatial awareness similar to existing algorithms such as those used by Google Maps.

The key contributions of this work is the design and release of a new benchmark to assess geospatial reasoning capabilities.

**Strengths:**

- The paper proposes a new benchmark to assess geospatial reasoning capabilities of llms
- The benchmark is created with diverse range of trajectories from real world coordinates, and tries to reduce data imbalance and biasness.
- This can be a good addition to existing llm benchmark suites and provide a unified evaluation framework.
- The paper is clearly written and explains in detail the steps used for creating the benchmark and evaluating models.

**Weaknesses:**

- The authors claim that there have not been previous works that evaluate the ability of llms to plan paths or read road networks or generate coordinates. This is false as there are multiple existing works in this domain such as (https://arxiv.org/pdf/2306.00020) who have already shown that llms contains a strong degree of groundness to real world data and can reason and plan using geospatial data.
- The benchmark only evaluates the capability of llms in using their internalized knowledge and world coordinates. I am not sure how this might be useful in real world applications, as one can easily call an external api or service to convert between locations and coordinates. Moreover llms are still not reliable in generating very accurate and fine floating point values such as longitude and latitude coordinates, as exhibited by the high MAE scores. The benchmark can be more useful if it addresses how language agents with access to external tools might perform in this scenario.
- Being able to recover a trajectory does not guarantee a high degree of understanding or reasoning capability. It shows that the llm was trained with a lot of geospatial data and can retrieve that knowledge given the correct prompt. Tasks such as RQ4 are more relevant to understanding whether the llm can actually use the knowledge and apply it to solve other tasks such as personalized recommendation. I think the authors can improve the benchmark much more by designing tasks such as these that require multi step reasoning, rather just recalling geospatial coordinates from memory.

**Questions:**

The paper addresses most of the relevant questions.

---

> ### Author Response · Authors · 2025-11-17
> **Response to Reviewer pXYj**
>
> Thank you for the constructive feedbacks. We address the concerns below.
>
> **Previous works exploring LLMs capabilities to do geospatial tasks**
>
> A: We stress that the novelty of this work is in the formulation, not the task itself. For example, in https://arxiv.org/pdf/2306.00020, the planning is done by providing location name (e.g., Broadway Theater in NYC to Hilton Hotel), and the output is textual description of the navigation step. From these the authors then manually map them to a map. Regarding other tasks that use GPS coordinates, they are mainly to test the LLM knowledge about the location (such as geocoding, predicting location given a coordinate, or drawing the outline of countries). In our task, we evaluate trajectory reconstruction, which requires sequential coordinate generation and the ability to plan by reading a road network, hence spatial reasoning, not just LLMs’ internal knowledge about geography.
>
>
> **Only test LLM's internal knowledge, no tool use**
>
> A: We did not only evaluate the capability of LLMs in using their internalized knowledge and world coordinates. We in fact first retrieve a road network surrounding the traces and then test the ability of the models to reason over this road network to form a valid path, taking into account the context of movement (such as modes of transportations, speed, and distance of travel). Further, we ensure accurate generation of fine floating-point values in Stage 2 of the framework. Using agents is a valid extension but is not the intended purpose of this paper as we position the benchmark to test the geospatial reasoning ability of LLMs.
>
> **The task only test LLM's ability to recall from memory**
>
> A: We intentionally construct this benchmark using traces from 2024 and 2025 onward, ensuring that data contamination is minimal. The Cutoff Analysis (Appendix J) shows no difference in how models perform beyond their cutoff date. The two-stage framework also shows that the LLMs are not simply "recalling geospatial coordinates from memory" as the models first need to generate navigation steps based on using the provided data from the road network.
> Thank you for your suggestion regarding extending the preference-based dataset. This is of course a very good direction but again out-of-scope.It is worth another paper on its own.

---

> ### Author Response · Authors · 2025-11-25
>
> Thank you for your time in engaging in the response period. As the rebuttal deadline is approaching, we would like to check if there are any remaining concerns or questions we could help clarify. We remain open to further discussion.

---

### Official Review · Reviewer_zLdE · 2025-11-02

**Soundness:** 2
**Presentation:** 2
**Contribution:** 2
**Rating:** 2
**Confidence:** 4

**Summary:**

This paper explores whether LLMs can perform geospatial reasoning by reconstructing masked GPS trajectories from road network data. It introduces GLOBALTRACE, a diverse dataset of 4K real-world trajectories, and proposes a two-stage prompting framework for path selection and coordinate generation. Experiments show that strong LLMs outperform specialized trajectory models and approach Google Maps’ performance, revealing both reasoning ability and regional biases. The paper also showcases preference-aware navigation scenarios, highlighting new applications of LLM-based spatial reasoning.

**Strengths:**

- Introduce a new task for geospatial reasoning via trajectory recovery
- Constructs a comprehensive dataset (GLOBALTRACE) across regions and transportation modes
- Design a prompts to tackle
- Comp[are with LLM and non LLM HMM, Google Maps, TrajFM, etc.)
- Clear metric design (MAE, PoT, F1 variants) with detailed ablations

**Weaknesses:**

- This work lacks the clear motivation and its impact, other than can LLM do this? And how is it distinct from path finding or routing or navigation given source and destination in term of consequence?
- No visual or multimodal grounding—limited to text-based reasoning. While the application seems valuable, the current text form is not much, rather I think a multi-modal task would be more significant.
- The benchmark is seems easy, not challenging, on par the Google maps-which limits its applicability. No clear motivation why to use LLM instead of these methods. Also High token/inference cost limits real-world scalability but there was no study analysis.
- No discussion on human performance as well.
- Section seems not method rather evaluation protocol.
- Data collection section with details is missing
- For Preference aware path finding analyis, Popular path finding is a common task, why not sudy on that?
- It does not discuss why or how the two stage prompt is derived, neigher consider any other reasoning prompts.
- I dont understand why the dataset is based on OpenStreetMap and the evaluation consider Google maps.
- Bias discussion shallow—not linked to data imbalance quantitatively
- Missing robustness tests on noisy or incomplete maps or any such discussion
- Not much intuition on error corrections, why it was making the error and how to fix.

**Questions:**

See weakness

---

> ### Author Response · Authors · 2025-11-17
> **Response to Reviewer zLdE (1/2)**
>
> Thank you for your constructive feedbacks. We address the concerns below:
>
> **Motivation, impact, and how the difference with common navigation problem**
>
> A: Spatial reasoning is a core component of general intelligence. Our benchmark provides a valuable resource to systematically evaluate this capability in LLMs. We select trajectory recovery as a task to evaluate this because this is a long-horizon task, which requires models to maintain consistency in its generation rather than a single prediction:
> - Unlike previous trajectory recovery works where the goal is just to predict a single point in between given points before and after, we want the model to generate a sequence of points that form a valid path.
> - We reconstruct how someone actually moved by using real user traces, requiring models to reason about transportation modes and speed patterns (See Section 3). We also force the model to reason over a road network covering a large area surrounding the traces to test its ability to read and understand maps, not just recalling from internal knowledge. This is crucial as it shows adaptability to new areas/regions (also why we want to ensure global coverage with popular/underrepresented regions).
> - As demonstrated in Section 6.4, LLMs' ability to comprehend road networks enables preference-aware navigation that goes beyond shortest-path routing. This opens applications in personalized navigation, accessibility planning, and urban mobility analysis that traditional systems cannot address.
>
> **Visual and multimodal**
>
> A:
>
> - In a way, graph data (road network retrieved from OSM) are also another modality (they are JSON text with structure and metadata, not natural language). Structured map data is also the standard data source used in map-based navigation systems.
> - As for visuals, current VLMs still lag behind LLM in terms of capabilities, especially for tasks that require high precision, such as counting and reading clock, let alone reading maps with dense elements (POIs, road names, intersections, or directions of the roads) . A simple interaction in ChatGPT can confirm this (attach an image of a map and ask the model to draw a path connecting two points). The visual component would be helpful in more controlled environments such as indoor navigation (with the layout of a shopping mall or house).
>
> **Benchmark is easy, token cost analysis**
>
> A: Google Maps performance results are still relatively low (~65% PoT), so the benchmark is not  easy. The results also show that real user trajectories often deviate from optimal paths, making this inherently challenging (Section 6.1, Table 1). Specialized models for trajectory recovery also fail completely: TrajFM achieves only 15.3% PoT despite being designed for trajectory recovery, demonstrating non-trivial difficulty. We also discuss how this task differs from previous trajectory recovery datasets in multiple places in the paper (Intro, Problem Statement, and Related Work) . We did include token count analysis in the ablation study (Table 3, detailing the average number of tokens for each setting. From this, one can easily derive the cost based on their selected models and infrastructure.
>
> **Human performance**
>
> A: We acknowledge this limitation. Human evaluation would require extensive annotation effort given the 4,095 trajectories and open-ended nature of movement (multiple paths to reach a location). We prioritized comprehensive automated evaluation across multiple models and settings. On the other hand, humans mostly rely on Google Maps so in a way we can consider Google Maps to be the upper-bound in terms of performance.
>
> **Section seems not method rather evaluation protocol**
>
> A: Can you elaborate more? If regarding Section 4, we structured it to explain our two-stage framework (path selection → coordinate generation) which is our core methodological contribution.
>
> **Data collection details**
>
> A: Data collection details are provided in Section 3 and Appendix B. Can you elaborate what part of the data collection process is unclear?
>
> **Reason for two-stage setup**
>
> A: Our two-stage design emerged from empirical exploration (Section 6.2):
> - Direct approaches failed due to information overload (Table 3)
> - Decomposition into planning (Stage 1) and execution (Stage 2) mirrors human navigation and can facilitate more detailed qualitative evaluation.
>
> **OpenStreetMap rather than Google Maps**
>
> A: OpenStreetMap is a data source. We use OpenStreetMap because it is publicly available, and would facilitate reproducibility to compare a wide range of methods. Please note that Google Maps API is a method, and the input we provide to the API is just GPS coordinates, it will still use its own Google map data to do the routing. Also, we chose Google Maps API following previous work (https://openreview.net/forum?id=hS2Ed5XYRq).
>
> (cont.)

---

> ### Author Response · Authors · 2025-11-17
> **Response to Reviewer zLdE (2/2)**
>
> (cont.)
>
> **Bias discussion**
>
> A:  We provide quantitative bias analysis with a subset sampled to maintain balance between regions, activities, and gap length:
>
> - Regional disparities up to 50% between Global North/South (Figure 4)
> - Activity-specific performance gaps (Table 12)
> - Temporal analysis showing no data contamination (Figure 7)
>
> **Missing robustness tests**
>
> A: While not explicitly adding noise to the samples, our data is collected from real user traces and would inherently contain natural GPS noises in them. Future work could systematically study degraded map quality. Given the scope of the paper, this is a nice-to-have experiment, and would not add to the core findings for which we claimed.
>
> **Not much intuition on error corrections**
>
> A: We provide error analysis through:
>
> - Stage-based metrics showing where failures occur (Table 2)
> - Directional analysis revealing precision vs. recall trade-offs (Tables 9-11)
> - Case studies comparing LLM vs Google Maps errors (Figure 6)

---

> ### Author Response · Authors · 2025-11-25
>
> Thank you for your time in engaging in the response period. As the rebuttal deadline is approaching, we would like to check if there are any remaining concerns or questions we could help clarify. We remain open to further discussion.

---

### Author Response · Authors · 2025-11-18
**General comment**

We thank the reviewers for their efforts in reviewing the paper. While reviewers raise valid points, we believe many stemmed from misunderstanding the tasks.

**First, most reviewers question the motivation for this work and why using LLMs**. We want to stress again that the main motivation for this work is on evaluating the geospatial reasoning capabilities of LLMs, and from that we explore broader impacts of the main finding that they can comprehend road network topology and understand GPS coordinate well (thus, the use case on preference-based navigation).

**We are not asking "Can LLM replace traditional navigation systems?" but rather "What are the geospatial reasoning capabilities of LLM?"** and chose trajectory recovery as the task to explore this question as it is a challenging task and can show multiple aspects of geospatial reasoning capabilities. We described the task and how they differ from both standard trajectory recovery tasks (where the target is usually just to predict a single point) and routing tasks (where the output are textual description of the navigation steps, usually between named locations such as some places of interest). Above all, spatial intelligence is an important component of AGI. Our challenging benchmark provides a valuable resource to evaluate this capability. We also discuss a use case on how the finding of this work has broader impact in providing a more flexible way to enhance traditional routing (by integrating personal preferences).

While presentations and contributions rating are up to reviewers perception and we totally respect that, we believe the soundness ratings (1, 1, 2) appear to be caused by three issues:

- **Missing cost analysis**: We have reported the token count costs. One can derive the inference cost based on the actual costs of their chosen LLMs..
- **Misunderstanding of metrics and baselines**: Reviewers interpreted 60% PoT as "coin flip" when it actually represents substantial path reconstruction. Also, the reviewers misunderstood the task formulation, leading to confusion regarding the Google Maps API baseline, as well as assuming maps to be images. This raises a concern on the confidence scores given by the reviewers. .
- **Prompt sensitivity**: Most latest LLMs are not heavily  sensitive to wording. Instead, we conduct ablation studies on different ways to represent the context (i.e., the road network) and how it affects the model performance, which is a more  important part that needs to be dissected rather than the change of wording.

We hope this general comment, along with the individual response to each reviewer would make the contributions clearer.

---

### Note · Authors · 2026-01-05

**Comment:**

After careful consideration, we decided to withdraw the paper to refine the writing to make the contribution clearer.

**Withdrawal Confirmation:**

I have read and agree with the venue's withdrawal policy on behalf of myself and my co-authors.